



# Wildfires in Northern Eurasia affect the budget of black carbon in the Arctic. A 12-year retrospective synopsis (2002–2013).

**N. Evangeliou [1, 2] *, Y. Balkanski [1], W. M. Hao [3], A. Petkov [3], R. P. Silverstein [3], R. Corley [3], B. L. Nordgren [3], S. P. Urbanski [3], S. Eckhardt [2], A. Stohl [2], P. Tunved [4], S. Crepinsek [5, 6], A. Jefferson [6], S. Sharma [7], J. K. Nøjgaard [8], H. Skov [8]**

[1] CEA-UVSQ-CNRS UMR 8212, Laboratoire des Sciences du Climat et de l'Environnement (LSCE), Institut Pierre et Simon Laplace, L'Orme des Merisiers, F-91191 Gif sur Yvette Cedex, France.

[2] Norwegian Institute for Air Research (NILU), Department of Atmospheric and Climate Research (ATMOS), Kjeller, Norway.

[3] Missoula Fire Sciences Laboratory, Rocky Mountain Research Station, United States Forest Service, Missoula, Montana, USA.

[4] Department of Applied Environmental Science, Stockholm University, Stockholm, Sweden

[5] Cooperative Institute for Research in Environmental Sciences, University of Colorado, Boulder, Colorado, USA.

[6] NOAA Earth System Research Laboratory Physical Sciences Division/Polar Observations & Processes, Boulder, Colorado, USA.

[7] Climate Research Division, S&T Branch, Environment Canada, Toronto, Ontario, Canada.

[8] Department of Environmental Science, Aarhus University, DK-4000 Roskilde, Denmark.

Correspondence to: N. Evangeliou (Nikolaos.Evangeliou@nilu.no)



**Abstract**
In recent decades much attention has been given to the Arctic environment, where climate
change is happening rapidly. Black carbon (BC) has been shown to be a major component of
Arctic pollution that also affects the radiative balance. In the present study, we focused on
how vegetation fires that occurred in Northern Eurasia during the period of 2002–2013
influenced the budget of BC in the Arctic. For simulating the transport of fire emissions from
Northern Eurasia to the Arctic, we adopted BC fire emission estimates developed
independently by GFED3 (Global Fire Emissions Database) and FEI-NE (Fire Emission
Inventory - Northern Eurasia). Both datasets were based on fire locations and burned areas
detected by  MODIS (MODerate resolution Imaging Spectroradiometer) instruments on
NASA's (National Aeronautics and Space Administration) Terra and Aqua satellites.
Anthropogenic sources of BC were estimated using the MACCity (Monitoring Atmospheric
Composition & Climate / megaCITY - Zoom for the ENvironment) emission inventory.
During the 12-year period, an average area of 250,000 $km^2$ $yr^{-1}$ was burned in Northern
Eurasia and the global emissions of BC ranged between 8.0 and 9.5 Tg $yr^{-1}$. For the BC
emitted in the Northern Hemisphere, about 70% originated from anthropogenic sources and
the rest from biomass burning (BB). Using the FEI-NE inventory, we found that 102±29 kt $yr^{-1}$
BC from biomass burning was deposited on the Arctic (defined here as the area north of
67ºN) during the 12 years simulated, which was twice as much as when using MACCity
inventory (56±8 kt $yr^{-1}$). The annual mass of BC deposited in the Arctic from all sources
(FEI-NE in Northern Eurasia, MACCity elsewhere) is significantly higher by about 37% in
2009 to 181% in 2012, compared to the BC deposited using just the MACCity emission
inventory. Deposition of BC in the Arctic from BB sources in the Northern Hemisphere thus
represents 68% of the BC deposited from all BC sources (the remaining being due to
anthropogenic sources). Northern Eurasian vegetation fires (FEI-NE) contributed 85% (79–
91%) to the BC deposited over the Arctic from all BB sources in the Northern Hemisphere.
Arctic total BC burden showed strong seasonal variations, with highest values during the
Arctic Haze season. High winter–spring values of BC burden were caused by transport of BC
mainly from anthropogenic sources in Europe, whereas the peak in summer was mainly due
to the fire emissions in Northern Eurasia. BC particles emitted from fires in lower latitudes
(35°N–40°N) were found to remain the longest in the atmosphere due to the high release
altitudes of smoke plumes, exhibit tropospheric transport resulting in a high summer peak of
burden, and grow by condensation processes.





In regards to the geographic contribution on the deposition of BC, we estimated that
about 46% of the BC deposited over the Arctic from vegetation fires in Northern Eurasia
originated from Siberia, 6% from Kazakhstan, 5% from Europe, and about 1% from
Mongolia. The remaining 42% originated from other areas in Northern Eurasia. For spring
and summer, we computed that 42% of the BC released from Northern Eurasian vegetation
fires was deposited over the Arctic (annual average: 17%). Vegetation fires in Northern
Eurasia contributed to 14% to 57% of BC surface concentrations at the Arctic stations (Alert,
Barrow, Zeppelin, Villum, and Tiksi), with fires in Siberia contributing the largest share.
However, anthropogenic sources in the Northern Hemisphere remain essential, contributing
29% to 54% to the surface concentrations at the Arctic monitoring stations. The rest
originated from North American fires.



## 1 Introduction

The Arctic environment has experienced rapid modifications (e.g. warming, ice degradations, etc.) during the last four decades and concerns have been raised that human activities were the main cause for these transformations. The thinning of Arctic sea ice (Hansen and Nazarenko, 2004) and the Arctic's rapidly growing human influence (e.g. transportation, drilling, industry) indicates the need not only for further decrease of greenhouse emissions, but also a better understanding of aerosol properties, as well as of aerosol interaction with radiation, clouds, and ecosystems in Polar Regions. The "Arctic Haze" phenomenon in winter and spring is a major feature of Arctic air pollution. Several studies have been conducted to determine the sources of Arctic air pollution using trajectory, regional, and global models (e.g., Hirdman et al., 2010a; Klonecki et al., 2003; Koch and Hansen, 2005; Law and Stohl, 2007; Stohl, 2006). They all agree that the majority of the pollution in the high-latitude Arctic, especially near the surface, originates at mid- and high-latitudes, and that the accumulation of pollution in the Arctic is a consequence of the slow removal processes in winter and spring (Shaw, 1995). Also, Northern Eurasia (Europe, Siberia, Kazakhstan, Mongolia, etc.) is the main source of the Arctic BC due to both wildfire and anthropogenic emissions.

Episodic emissions from mid- and high-latitude vegetation fires can affect tropospheric concentrations of trace gases (e.g. carbon monoxide (CO), ozone ($O_3$), volatile organic compounds (VOC), and aerosols (e.g. BC)) several thousand kilometers away from the sources (Forster et al., 2001; Wotawa and Trainer, 2000). Additionally, emissions from boreal fires lifted by convection can substantially alter upper tropospheric and the lowermost stratospheric radiation balance and chemistry (Waibel et al., 1999; Jost et al., 2004; Fromm et al., 2005). Aerosols and trace gases are uplifted during transport to the Arctic due to the upward sloping surfaces of constant potential temperature towards the Arctic (Klonecki et al., 2003; Stohl, 2006). However, understanding of aerosol transport from mid-latitudes to the Arctic has been limited because of the lack of quantification of the relevant aerosol sources and removal processes.

Globally, BC contributed to climate warming with recent estimates of radiative forcing at the top of the atmosphere ranging between 0.17 and 0.71 W m$^{-2}$ (Bond et al., 2013; Myhre et al., 2013; Wang et al., 2014). Snow albedo may be reduced by 1–3% in fresh snow by BC deposited in the Arctic and by another factor of 3 as snow ages and the BC becomes more concentrated (Clarke and Noone, 1985). Hansen and Nazarenko (2004) found that the decreased albedo in Arctic snow and ice since preindustrial times resulted in a hemispheric





radiative forcing of +0.3 W m$^{-2}$, which may have had a substantial impact on the climate in
the Northern Hemisphere, while for Northern Russia it amounts to 0.2 W m$^{-2}$ (Lee et al.,
2013a; 2013b). Airborne soot also absorbs incoming solar radiation thus warming the air and
reducing tropical cloudiness (Ackerman et al., 2000).

5       Biomass burning (BB) constitutes a major source of BC, in addition to incomplete

combustion of fossil fuels (primarily coal and diesel) and burning of biofuels. BC is the most
absorbing portion of carbonaceous aerosols, commonly referred to as "soot". Vegetation fires,
either anthropogenic or natural, constitute a major source of atmospheric BC concentrations.
BC deposited on snow/ice surfaces reduces surface reflectance and can promote faster melting
of snow/ice in the Arctic, which is tightly coupled to climate effects through snow-albedo
feedback (Flanner et al., 2007, 2009; Hansen and Nazarenko, 2004). In addition, high aerosol
concentrations in the Arctic Haze lead to the enhancement of cloud longwave emissivity
(Garrett and Zhao, 2006; Lubin and Vogelmann, 2006), leading to surface warming and
accelerating the melting of snow/ice.

15       Model simulations by Stohl (2006) suggested that the contributions from BB to Arctic

BC loadings, particularly from fires in Siberia, exceeded the anthropogenic contributions in
the summer. Moreover, large amounts of BC from Siberia and Kazakhstan have been
observed during aircraft campaigns over Alaska in spring 2008 (Warneke et al., 2009), which
was a year with an unusually early start of the BB season in Northern Eurasia. Warneke et al.
(2010) estimated that BB in Russia may have doubled aerosol concentrations in the Arctic
Haze during the spring. BC has been monitored at several surface stations in the Arctic (e.g.
Alert in Canada, Barrow in Alaska, and Janiskoski in Russia) for many years (e.g. Sirois and
Barrie, 1999; Sharma et al., 2006; Quinn et al., 2008; Eleftheriadis et al., 2009; Gong et al.,
2010; Huang et al., 2010 and many others), showing decreasing trends during the 1980s and
1990s, which have been attributed to reductions in anthropogenic emissions (Sharma et al.,
2013; Hirdman et al., 2010b).

In this study, we focused on the transport of BC produced by vegetation fires in

Northern Eurasia to the Arctic from 2002 to 2013. We define Northern Eurasia from 10°W to
170°E and 35°N to 80°N. The Arctic is defined here as the area north of the Arctic Circle
(~67°N). Fires were mapped using satellite measurements from MODIS on NASA's Terra
and Aqua satellites. We investigated the geographic distribution of BC sources contributing to
the Arctic BC budget, which needed to be better understood for developing BC mitigation
policies. Moreover, the transport of BC to the Arctic after vegetation fires was defined as
transport efficiency of BC. Shindell et al. (2008) showed large differences in the calculated



BC concentrations in the Arctic using different General Circulation Models (GCMs). A large
part of these differences was attributed to the different model treatment of BC aging from
hydrophobic to hydrophilic and rainout/washout processes during transport. It indicated the
necessity of a continuously improving description of BC transport in order to better assess the
impact on the Arctic climate despite many recent improvements (e.g. Browse et al., 2012;
Eckhardt et al., 2015).
This paper consists of five sections. The methodology (transport model, model set-up,
emission altitude, satellite-derived BC emissions) is discussed in detail in the next section.
The results, with respect to Arctic transport and deposition of BC, are presented in Section 3.
Then, we show how different regions in Northern Eurasia contribute to the Arctic BC,
distinguishing between anthropogenic and BB sources (Section 3.3). In Section 4.1, we
discuss how our modeling results compare to observations of BC using data from five
different Arctic stations for the period of our simulations (2002–2013). Finally, we calculate
and study transport efficiencies of BC to the Arctic from different BB regions (Section 4.2).
The main conclusions are presented in Section 5.
**2  Methodology**
**2.1  The LMDz-OR-INCA model**
We used the LMDz-OR-INCA global chemistry-aerosol-climate model, which couples
the LMDz (Laboratoire de Météorologie Dynamique) GCM (Hourdin et al., 2006) and the
INCA (INteraction with Chemistry and Aerosols) model (Hauglustaine et al., 2004). The
interaction between the atmosphere and the land surface was ensured through the coupling of
LMDz with the ORCHIDEE (ORganizing Carbon and Hydrology In Dynamic Ecosystems)
dynamical vegetation model (Krinner et al., 2005). In the present configuration, the model
included 39 hybrid vertical levels extending to the stratosphere, and a horizontal resolution of
1.29°×0.94° (280 grid-cells in longitude, 192 in latitude). A more detailed description and an
extended evaluation of the GCM can be found in Hourdin et al. (2006). The large-scale
advection of tracers was calculated based on a monotonic finite-volume second-order scheme
(Hourdin and Armengaud, 1999). Deep convection was parameterized according to the
scheme of Emanuel (1991). The turbulent mixing in the planetary boundary layer (PBL) was
based on a local second-order closure formalism. A comparison made with inert tracers
indicated an enhanced vertical transport as the horizontal resolution of the model was
increased from 144×142 grid-points to 280×192.



The INCA model simulates the distribution of anthropogenic aerosols such as sulfates,
nitrate ($NO_3$), BC, particulate organic matter (POM), as well as natural aerosols such as sea-
salt and dust. The aerosol model keeps track of both the number and the mass of aerosols
using a modal approach to treat the size distribution, which is described by a superposition of
5 log-normal modes (Schulz, 2007), each with a fixed spread. To treat the optically relevant
aerosol size diversity, particle modes were categorized in three ranges: sub-micronic
(diameter < 1 μm) corresponding to the accumulation mode, micronic (diameter 1–10 μm)
corresponding to coarse particles, and super-micronic or super coarse particles (diameter > 10
μm). Compared to a bin-scheme, the treatment of the size distribution with modes was
computationally much more efficient (Schulz et al., 1998). Furthermore, to account for the
diversity in chemical composition, hygroscopicity, and mixing state, we distinguished
between soluble and insoluble modes. In both sub-micron and micron size ranges, soluble and
insoluble aerosols were treated separately. Sea-salt, $SO_4$, and methane sulfonic acid (MSA)
were treated as soluble components of the aerosol, dust was treated as insoluble, whereas
nitrate, BC, and POM appeared both in the soluble and insoluble fractions. The aging of
primary insoluble carbonaceous particles transfers insoluble aerosol number and mass to
soluble ones with a half-life of 1.1 days (Chung et al., 2002). The deposition scheme used in
the model is described in detail in Evangeliou et al. (2013).
**2.2   Model set-up, BC inventories and injection height**
The simulations lasted from January 1[st], 2002 to December 31[st], 2013. The model ran in
a nudged mode using 6-hourly ERA Interim Re-analysis data (ECMWF, 2014) with a
relaxation time of 10 days (Hourdin and Issartel, 2000).
To support the IPCC-AR5 (Intergovernmental Panel for Climate Change Assessment
Report 5) and the ACCMIP (Atmospheric Chemistry and Climate - Model Intercomparison
Project), a historical emissions dataset was developed (Lamarque et al, 2010) on a decadal
basis (from 1850 to 2000 for the historical dataset), as well as RCP (Representation
Concentration Pathways) emission scenarios for the period after the year 2000. As part of a
project funded by the European Commission, MACC (Monitoring Atmospheric Composition
& Climate) and CityZen (megaCITY - Zoom for the ENvironment), the ACCMIP and the
RCP emissions dataset were adapted and extended to the year 2013 on a yearly basis. For
anthropogenic emissions, emission data was interpolated on a yearly basis. For BB emissions,
the ACCMIP dataset was extended to yearly and monthly mean calculated from modified
RETRO (REanalysis of the TROposhperic chemical composition) BC emission data for the



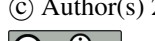

years 1980 to 1996, and from GFEDv3 carbon emission data for the years 1997 to 2013, by
applying a single set of vegetation type specific emission factors, and the predominant
vegetation map used in the GFEDv3 inventory. This extension of the ACCMIP and RCP
emission dataset for the MACC and CityZEN projects is referred to as MACCity
(MACC/CityZen) emission dataset (Granier et al., 2011). As it is explained below, emissions
of MACCity (anthropogenic and BB) were used as the input source of the model worldwide,
in which BB sources were derived from GFEDv3. In addition, we adopted BC emissions from
BB in Northern Eurasia from 2002 to 2013 (FEI-NE) described in the companion paper (Hao
et al., 2015), while MACCity emissions were used for all the sources in other regions and for
anthropogenic sources in Northern Eurasia.

11       Injection height is a key factor that controls transport and in turn deposition of BC

emitted from fires. It is generally accepted that only explosive volcanic eruptions and strong
crown fires (more common in North America than in Northern Eurasia) have the energy to
inject pollutants from the surface into the stratosphere (Jost et al., 2004; Fromm et al., 2005).
In a modeling study of mid-latitude supercell thunderstorms (Wang, 2003), it was reported
that these plumes could induce important transport into the lowermost stratosphere. These
findings suggest that extreme convection, even unassociated with energetic forest fires, may
represent an important pathway for rapid, efficient redistribution of gases and particles from
the lowest levels of the atmosphere to the lower stratosphere or, more commonly, the upper
troposphere. Nedelec et al. (2005) described such a case for a fire happening over Siberia.
However, injection of emissions in the lower troposphere is more common. Recently, Sofiev
et al. (2013) published global maps of emission heights of wildfires that occurred between
2000 and 2012 reporting that about 80% of the smoke is generally injected within the PBL,
while the rest is injected at higher altitudes. Here we follow the same pattern as Sofiev et al.
(2013) for Northern Eurasia, where 90% of the emissions were injected below 1.1 km, while
the rest in heights up to 1.5 km.
**2.3   BC emissions from FEI-NE and MACCity**

For the 12-year global simulations, anthropogenic sources of BC were adopted from the

MACCity emission database. As regards to BB emissions, MACCity BB emissions from
GFED3 were applied for all the regions outside Northern Eurasia, while within Northern
Eurasia anthropogenic emissions from MACCity and BB from FEI-NE (Hao et al., 2015)
were adopted. In summary, BC emissions from BB, excluding agricultural fires, in Northern





Eurasia were estimated based on the area burned, fuel loading, percentage of the fuel burned,
and emission factors of BC from different vegetation types (Hao et al., 2015).

3        This combined simulation is hereafter referred to as FEI-NE+MACCity. For

comparison, we carried out the same simulation but using MACCity emissions alone (i.e. BB
emissions in Northern Eurasia were also taken from MACCity) for the same period (2002–
2013) (from now on referred to it as MACCity simulation). The different simulations are
shown in Table 1.
**2.4    Lifetime calculations**

9        Several definitions for atmospheric lifetime exist. In any domain of Earth's atmosphere

the mass balance can be expressed as:
$\frac{dB(t)}{dt} = S(t) - \frac{B(t)}{\tau(t)}$                                                                (Eq. 1)
where $B(t)$ is the atmospheric burden, $S(t)$ is the mass entering or exiting the domain and
$\tau(t)$ is the removal time over a given time-step. If one assumes equilibrium between $S(t)$ and
deposition (steady state conditions), the mean steady state lifetime will be:
$\tau_{ss} = \frac{\overline{B}}{\overline{D}}$                                                                          (Eq. 2)
where $\overline{B}$ and $\overline{D}$ are the mean atmospheric burden and deposition over a specific period (Croft
et al., 2014).
**2.5    Observation data**

19       With different types of instruments, we  collected measurements of BC, which may not

always be directly comparable. Following the nomenclature of Petzold et al. (2013), we
referred to measurements based on light absorption as equivalent BC (eBC) and
measurements based on thermal-optical methods as elemental carbon (EC).

23       Aerosol light absorption data was obtained from five sites in different parts of the

Arctic: Alert, Canada (62.3°W, 82.5°N; 210 m above sea level (a.s.l.)), Barrow, Alaska
(156.6°W, 71.3°N; 11 m a.s.l.), Zeppelin/Ny Ålesund, Spitsbergen, Norway (11.9°E, 78.9°N;
478 m a.s.l.), and Tiksi, Russia (128.9°E, 15 71.6°N; 1 m a.s.l.). Different types of particle
soot absorption photometers (PSAPs) were used for the measurements at Barrow and
Zeppelin, and an aethalometer was used at Alert and Tiksi. All these instruments measured
the particle light absorption coefficient $\sigma_{ap}$, each at its own specific wavelength (typically at
around 530–550 nm), and for different size fractions of the aerosol (typically particles smaller
than 1, 2.5, or 10 μm are sampled at different humidities). Conversion of $\sigma_{ap}$ to eBC mass



concentrations is not straightforward and requires certain assumptions (Petzold et al., 2013).
The mass absorption efficiency used for conversion can be specific to a site and is uncertain
by at least a factor of two. For Tiksi, the conversion was done internally by the aethalometer.
For the other sites, a mass absorption efficiency of 10 m$^2$ g$^{-1}$, typical of aged BC aerosol
(Bond and Bergstrom, 2006), was used. Sharma et al. (2013) used an even higher value of 19
m$^2$ g$^{-1}$ for Barrow and 10 m$^2$ g$^{-1}$ for Alert data.

7        At Villum Research Station, Station Nord, Greenland, thermo-optical measurements

were available. Weekly aerosol samples were analyzed with a thermal-optical Lab OC/EC
instrument from Sunset Laboratory Inc (Tigard, OR, USA). Punches of 2.5 cm$^2$ were cut from
the filters sampled at Villum and analyzed according to the EUSAAR-2 protocol (Cavalli et
al., 2010).

12        At Alert, eBC data was available for the years 2006–2013, at Barrow for 2002–2013, at

Tiksi for 2009–2013, and at Zeppelin for 2002–2013. At Villum station, EC data was
available for 2008–2011. The EC and eBC data were directly compared with modeled BC
concentrations for the same locations and periods. Tiksi data was not filtered for a clean air
sector and may have been affected by local pollution events. Barrow and Alert data were
routinely subject to data cleaning, which removed the influence from local sources. Zeppelin
generally was not strongly influenced by local emissions; however, summer values were
enhanced by some 11% due to local cruise ship emissions (Eckhardt et al., 2013).

## 20  3   Results

### 21  3.1   Emission, transport and deposition of BC

22        Northern Eurasia encompasses diverse ecosystems including forest, shrubland,

cropland, grassland, and savanna (Friedl et al., 2002). The total burned areas (excluding
agricultural fires) during the period of 2002–2013 were estimated to be 250,000 km$^2$ yr$^{-1}$ (n =
12) (Hao et al., 2015), which consisted of 61% of grassland and 27% of forest. Grassland fires
occurred predominantly over Central and Western Asia, and forest fires over Siberia in
Russia. The years 2003, 2006, and 2008 showed 96%, 40%, and 30%, respectively, more fire
detections than the annual mean from 2002 to 2013 (Figure S 1). The unusual high fire
activity in 2003, 2006, and 2008 was a result of extensive grassland fires over Central and
Western Asia, and forest and grassland fires over Russia.





Table 2 presents the yearly mean atmospheric emissions of BC from anthropogenic and
BB sources for the period 2002–2013 from the FEI-NE+MACCity and the MACCity
simulation. Annual global BC emissions varied in a range from 8.02 Tg in 2007 to 9.48 Tg in
2003 with an average: 8.42±0.43 Tg yr$^{-1}$ over the period 2002 to 2013 according to FEI-NE
and MACCity. These values compared well with other published results. For instance, Wang
et al. (2014) reported that, according to PKU-BC-2007 (Peking University BC Inventory for
2007) inventory of global BC emissions (both anthropogenic and biomass burning sources), a
total amount of 8.9 Tg was emitted in 2007. The ECLIPSE inventory estimated for 2010 by
Eckhardt et al. (2015) was 8.32 Tg, only slightly higher than our estimations of 8.02 Tg. In
contrast, the ACCMIP BC emissions for 2005 (Lamarque et al., 2010) was 15% lower (e.g.
~7.82 Tg compared to 8.13 Tg). BC emissions from vegetation fires in Northern Eurasia
ranged between 0.45 and 2.19 Tg for the period 2002–2013 representing 20% to 49% of the
total BC emissions over this region. In comparison, the emissions over the same region based
on the MACCity inventory ranged between 0.13 and 0.43 Tg and accounted for 5% to 16% of
total BC emissions. The BC emissions in the Northern Hemisphere (FEI-NE+MACCity) were
classified as 70% from anthropogenic sources and 30% from BB. Northern Eurasian
vegetation fires accounted for 56% of the BB BC emissions in the Northern Hemisphere
(Bond et al., 2013).
BC emissions from FEI-NE were relatively constant in time except for four intense
years (Figure S 2). Similarly, the GFED3 of MACCity showed relatively high emissions in
some of these years (2003 and 2006), but the relative emission increase was significantly
lower for these years (Table 2). The fire episodes that occurred in Northern Eurasia during
these extreme fire years were particularly intense. In 2003, fire events in the Transbaikal
region (Russian provinces Chita and Buryatia) caused severe smoke pollution in the Far East
of the Russian Federation (IFFN, 2004), while in spring 2006, smoke from peat and forest
fires in the western Russian Federation was noted as far as the United Kingdom (Hao et al.,
2009) and in the Arctic (Stohl et al., 2007). In summer 2006, smoke from vegetation fires in
the Russian Federation persisted for weeks over Finland (GFMC, 2006).
Figure 1 depicts deposition anomalies of BC for the period 2002–2013. A remarkable
feature was that the highest anomalies were observed in the northernmost part of Asia and the
Arctic in 2003, 2006, 2008, and 2012. This indicates that during these years the largest
amounts of BC were deposited over Arctic regions as a result of large fire events in Siberia,
Western Russia, and Kazakhstan. The annual amount of BC deposited over the Arctic from
all possible global sources (including BB) during the 12-year period ranged from 65 kt to 152





kt (average: 102±29 kt yr$^{-1}$) representing about 0.9–18% of total global emissions. The total
annual deposition of BC to the Arctic from vegetation fires in Northern Eurasia (FEI-NE)
during the same period ranged between 29–120 kt (average: 65±28 kt yr$^{-1}$) or about 0.3–14%
of total global BB emissions (Table 2). Hence, more than half of the total BC that deposited
over the Arctic originated from BB in Northern Eurasia and underlined the importance of the
Northern Eurasian vegetation fires on the Arctic BC budget, on par with the contribution of
other sources.
Figure 2 illustrates BC deposition from FEI-NE detected vegetation fires only,
excluding anthropogenic sources, while Figure S 3 shows the deposition from all sources
(FEI-NE+MACCity). For the most intense fire years, 2003, 2006, 2008, and 2012, an annual
amount of 142, 129, 117, and 152 kt of BC, respectively, was deposited over the Arctic from
Northern Eurasian vegetation fires (Table 2). It amounts to 3.2 to 4.1 times the 37±5 kt yr$^{-1}$
average (2002 to 2013) deposition flux for anthropogenic sources over Northern Eurasia.
Finally, when using MACCity emissions, both from anthropogenic and BB sources, the
estimated average deposition of BC over the Arctic was 56±8 kt yr$^{-1}$ for the period 2002–
2013. Consequently, Arctic deposition was lower by 45% compared to FEI-NE, when BB
emissions of BC from GFED (MACCity) were used.
**3.2  Aerosol lifetime and seasonality of BC**
Figure 3 depicts the global mean aerosol lifetimes for anthropogenic and BB BC from
the combined FEI-NE+MACCity simulation. The results are given in Box & Whisker plots of
daily lifetimes for the 12-year period. Mean aerosol lifetimes for anthropogenic BC were
stable for all the studied years with an annual average value of 5.6±0.2 d. Mean aerosol
lifetimes of BC from FEI-NE+MACCity simulation exceeded these lifetimes by 1.2 days
(6.8±1.0 d). The pronounced variability of the mean lifetime of BB BC was attributed to the
variability in regions and injection height. According to the injection scheme used (Sofiev et
al., 2013), continuous injection close to the PBL (around 65% of the mass of BC is emitted up
to 0.8 km, 90% up to 1.1 km) results in accumulation of BC in the troposphere, where longer
lifetime occurs compared to the PBL. This was likely the main reason for transport of BC into
the Arctic. For any soluble species emitted in the PBL, the lifetime is controlled by the
removal, which happens within a few days if it is not transported to the free troposphere. Two
weeks after being injected in the atmosphere, most of the tropospheric aerosol has been
scavenged through wet deposition, whereas aerosols that have been transported into the high
troposphere/lower stratosphere persist, given the absence of wet scavenging at these heights.





The total aerosol mass and the lifetimes are then dominated by the stratospheric loading
(Cassiani et al., 2013).
Mean aerosol lifetimes from global models are typically in the range of 3–7 days
(Benkovitz et al., 2004; Textor et al., 2006), very similar to our estimations for BC, but more
variable in this study due to the higher injection heights of BC in smoke plumes. As stated, in
the present case of wildfires in Northern Eurasia, the lifetimes and the behaviour of BC was
strongly affected by the fact that it was directly emitted aloft. Although 80–90% of BC was
emitted inside the PBL, nearly 65% of the BC was emitted near the PBL height, while 10%
was injected above (according to Sofiev et al., 2013).
In this study, we analyzed monthly values of both the Arctic BC burden and the mass
deposited in the Arctic for all BC sources (FEI-NE+MACCity) and for BC produced from BB
sources only over Northern Eurasia (FEI-NE). A strong seasonal variation can be seen in both
Arctic burden and deposition (Figure 4a, b, c and d). Relatively high BC burden from the
combined emissions of the FEI-NE+MACCity run occurred in winter (December–January,
Figure 4b), while two peaks were found in late spring and in summer for intense fire years.
The latest were clearly caused by spring and summer fire events over Northern Eurasia
(Figure 4). The higher winter values corresponded to the meridional transport of BC, mostly
emitted from anthropogenic sources (Figure 4a). These large increases in the burden were not
proportional among the different aerosol modes (hydrophilic and hydrophobic) and affected
mostly hydrophilic BC aerosol from vegetation fires. This indicated that wet scavenging was
less efficient than the rest of the year along the transport path between mid- and high-
latitudes. A spring maximum was also simulated for BC deposition over the Arctic. It was
caused by a combination of anthropogenic BC in the Northern Hemisphere and fires in
Northern Eurasia (Figure 4a and c), the majority of which occurred between March and May.
The annually deposited mass of BC over the Arctic exceeded the annual average for the
period 2002–2013 by 52% in 2003 and 86% in 2012.
There were three noteworthy findings. First, the year 2012 appeared to be a year when
BC transport to the Arctic was particularly efficient. Although the year itself (2012) was not
an extreme fire year (Figure S 1) in terms of fire detections, it appeared that the prevailing
winds and the lack of scavenging in mid-latitudes during June and July when fires were most
intense, favored transport and subsequent deposition to high-latitudes (120 kt from vegetation
fires only and 152 kt from all possible sources). Second, we simulated a higher relative
contribution of wet to total deposition in the Arctic (90%) than at mid-latitudes (69%). Third,





the annual mean lifetime of anthropogenic BC particles from BB was longer (6.8 d) than for
BC from combustion (5.6 d). These values are within the range of other published results
(e.g., 5.8 days from Park et al. (2005) and 7.3 days from Koch and Hansen (2005)).
**3.3   Geographic distribution of sources contributing to the Arctic BC**
In this section, we compare the contributions from different source regions and from
different emission types to the deposition of BC in the Arctic region. Several simulations with
BC tracers tagged by source region were carried out to isolate the different contributions. We
selected the following regions (Figure 5a): Europe, Asia, Siberia, Kazakhstan, and Mongolia,
as well as from the latitude bands 35°N to 40°N, 40°N to 50°N, 50°N to 60°N, above 60°N
(60°N–90°N), and also the entirety of Northern Eurasia. We discuss the results of these
separate simulations and compare them to our FEI-NE+MACCity run (Figure 5). Each
simulation covered the whole period from 2002 to 2013.
Quinn et al. (2007) were the first to report that for BC over the Arctic, anthropogenic
sources dominated during winter and early spring Arctic Haze conditions. We estimated that
anthropogenic emissions accounted for 70% of the Arctic BC burden from all sources in the
North Hemisphere during winter and fall and for 40% during spring and summer when
vegetation fires are most frequent.
Vegetation fires from Northern Eurasia contributed 68% of the annual BC deposition
over the Arctic coming from that same region (Figure 5d), whereas anthropogenic emissions
constituted a lesser share (32%). Northern Eurasian vegetation fires were the most numerous
ones over the Northern Hemisphere; they contributed for 81% of the Arctic deposition of BC
from North Hemispheric fires (Figure 5d), while the rest came from other sources (e.g. BB in
North America).
Of the Northern Eurasian BB BC deposition in the Arctic, 95% was from Asia, while
only 5% came from Europe. On a more regional basis, Siberia contributed 46% of the
Northern Eurasian BB BC deposited in the Arctic, whereas Kazakhstan contributed 6%, and
Mongolia only 1%. The rest was shared between fire events in Europe (5%) and elsewhere in
Asia (42%) from areas that were not masked.
The relative contributions of fires at different latitudes to the Arctic BB BC deposition
were distributed as follows: fires from the 35°-40°N latitudinal band over Northern Eurasia
contributed only 7%, from 40°-50°N the contribution was 21%, 40% of this deposition came
from fires at 50°-60°N, and 32% from fires above 60°N (Figure 5c).





Moreover, we examined the vertical distribution of BC over the Arctic for the different
source regions (Figure 6). When emitted from Europe, BC over the Arctic was found mostly
below 5 km (either in the PBL or the low free troposphere), while BC emitted from Asia was
found in higher layers that extended up to the mid- to high- free troposphere (Figure 6).
Similar vertical distribution for aerosols have been reported by Stohl et al. (2002), who
estimated that aerosol originating from Asia was mixed throughout the entire troposphere
within a few days.
These findings can be discussed in light of the ones reported by other authors. For
instance, our results agree well with those of Hirdman et al. (2010a), who reported that the
Northern Eurasian region (Europe and Russia) was the main contributor to the Arctic surface
concentrations of BC and sulfate. The present study shows the importance of all the regions
north of 50°N. Stohl (2006) reported that Asia contributed 10 times less than Europe to the
Arctic BC surface concentrations, which was not supported by our findings for total BC. Our
results agreed with the conclusions of Koch and Hansen (2005), who reported that Europe
contributed 10% to the Arctic deposition of BC (anthropogenic and BB). In any case, all
surface measurements of BC over the Arctic indicated that the main contributors to the Arctic
BC during the summer were high-latitude sources (Hirdman et al., 2010a; Sharma et al.,

18   2013).

**4    Discussion**
**4.1    Observed and simulated BC concentrations at Arctic surface stations**
There are contradictory results in the literature about where geographically the Arctic
BC pollution originates. For example, Koch and Hansen (2005) estimated that Southeast Asia,
Europe and Russia contributed each about 20–25% of the Arctic pollution during January to
March period, whereas Stohl (2006) reported that Europe was the main contributor for the
Arctic BC concentration at the surface. In addition, Huang et al. (2010) stated that Russia
contributed 67% to the surface Arctic pollution, whereas Europe and North America were
18% and 15%, respectively (16-year observations from Alert). Northern Eurasia appeared to
be the main contributor in terms of Arctic BC during most seasons (Hirdman et al., 2010a).
More recently, Stohl et al. (2007) and Warneke et al. (2009) reported that boreal and
agricultural fires in Eastern Europe and in Siberia might be strong contributors to Arctic BC
especially in the spring. Furthermore, Stohl et al. (2013) highlighted gas-flaring emissions in
high latitudes as a major contributor to the Arctic BC.





Figure 8 compares the simulated surface BC concentrations from this study with in-situ
eBC and EC measurements from monitoring stations at Alert (Canada), Barrow (Alaska,
USA), Villum (Greenland, Denmark), Zeppelin (Ny-Ålesund, Svalbard, Norway), and Tiksi
(Russian Federation). The observations, when available, were represented for the entire period
of our simulations (2002–2013). At Zeppelin and Barrow stations, the measurements were
available between 2002 and 2013, at Alert from 2005 to 2013, at Villum for 2008 to 2011,
and at Tiksi station for 2009 to 2013. Figure 7 compares the simulated versus observed daily
surface concentrations by a Box and Whisker plot at the five Arctic stations (Alert, Barrow,
Villum, Tiksi, Zeppelin) for the period 2002 to 2013.
At Alert, there was a systematic underestimation in winter and early spring and an
overestimation in late spring and summer, when the station did not record any enhanced eBC
concentrations that would be attributed to vegetation fires in Northern Eurasia (Figure 8). At
Barrow, the model not only accurately estimated surface concentrations during winter, when
the Arctic Haze is important, but also in spring and summer, when vegetation fires persisted
(Figure 8). At Villum, the measured seasonality was reproduced quite well by the model and
also individual episodes of elevated BC surface concentrations in spring and summer were
captured and attributed to vegetation fires. At the Tiksi station, the comparison showed a
notable deviation from the measurements. Although some peaks in spring and summer were
captured, the model missed completely the high concentrations of eBC in winter and early
spring. Finally, at Zeppelin, it appeared that the anthropogenic BC contribution was slightly
underestimated (spring), while vegetation fires in Northern Eurasia elevated modeled surface
concentration of BC.
Looking at other inventories and results from the Lagrangian particle dispersion model
FLEXPART, we are convinced that local anthropogenic sources play an important role in the
apparition of these peaks (Eckhardt et al., 2015). It was already noted that our model
underestimated surface concentrations at Alert and Villum stations during the Arctic Haze
period. This is likely the consequence of an underestimation of Arctic transport in the model
or a misleading emission inventory used for anthropogenic sources (MACCity). To verify it,
we estimated surface concentrations of BC in the same longitude as the five Arctic stations,
but five and ten degrees south in latitude (Figure S 4). In all stations except Tiksi, surface
concentrations increased to the south confirming that the underestimation by our model over
the Arctic could be attributed to a too weak transport simulated towards the Arctic.
It has been reported that most models underestimated BC in the Arctic during winter
and early spring (Eckhardt et al., 2015), likely due to an improper representation of the




scavenging processes (lack of below-cloud scavenging for solid phase water), the different
emission profiles used for BC, and underestimated emission inventories used as input to the
models (e.g. Koch and Hansen, 2005; Liu et al, 2011; Jiao et al., 2014). Here, it is apparent
that the BC concentrations over the Arctic are reasonable with the current model version
compared toobservations, with a tendency to underestimate winter concentrations and slightly
overestimate summer ones. The largest deviations occur at the station Tiksi, Russia. The
station is located only a few kilometers away from Tiksi town and local pollution is likely to
affect the measurements, as the town has both a small airport and a harbor. Despite these
drawbacks at Tiksi station, the data from the station has been used to estimate the sources in
Northern Eurasia (Cheng, 2014). Eckhardt et al. (2015) compared both surface- and aircraft
measurements of sulfate and BC in the Arctic to model output from eleven different models.
They found that the models generally underestimated the surface concentrations of BC and
sulfate in winter/spring, whereas concentrations in summer were overestimated. They also
found a strong correlation between surface measured sulfate and BC concentrations in
winter/spring (anthropogenic impact), which indicated that the sources contributing to sulfate
and BC were similar throughout the Arctic and that the aerosols were internally mixed and
undergo similar removal. Neither Eckhardt et al. (2015) nor Samset et al. (2013) could isolate
the reason to explain why some models performed better than the others.

19        In the present model configuration, we included emission inventory from FEI-NE and

MACCity's anthropogenic BC inside Northern Eurasia and MACCity (BB and
anthropogenic) outside Northern Eurasia (FEI-NE+MACCity), respectively, to evaluate if it
produced reliable results with respect to observations. The question that stemmed from this
comparison was whether or not existing datasets included all possible sources of BC
emission. This was examined by comparing FEI-NE+MACCity inventory with MACCity,
which included lower BC global emissions. Figure 8 depicts the difference of the average
atmospheric burden of BC between FEI-NE+MACCity and MACCity runs, while Figure S 5
and Figure S 6 show the same comparison for emissions and Arctic deposition of BC. It is
apparent that the difference in average atmospheric burden between FEI-NE+MACCity and
MACCity simulations (Figure 8) is positive over the Arctic showing that vegetation fires over
Northern Eurasia have a direct impact on the Arctic budget, especially during the most intense
fire years (2003, 2006, 2008, and 2012). The aforementioned impact extends up to North
America and may affect the BC concentrations there as well. Subsequently, the deviation of
the deposition of BC from Northern Eurasian vegetation fires relative to FEI-NE is shown to
be large over the Arctic (Figure S 6).





We also analyzed the influence of all anthropogenic and BB emissions from the regions
(defined in Table 1) to the average surface concentration of the Arctic stations (Figure 9). As
expected, the predominant contributor to the surface concentrations of the Arctic stations was
Northern Hemisphere anthropogenic emissions (29–55%) (e.g. Shaw et al., 2010). The
explanation is twofold. On one hand, transport of BC from the southern latitudes to the Arctic
takes place as the air-masses follow the trajectories of potential difference in temperature (a
dome effect), they are lifted up leaving the surface, especially from North America and Asia
(China). On the other hand, the transport from Russia/Siberia during the winter/spring is
closer to the surface due to large anthropogenic emissions that are also effectively transferred
from Europe. In addition, transport of BC from Russia/Siberia is less efficient during summer
due to pressure systems that block the BC transport. Fires from Northern Eurasia contributed
less BC to Barrow, Zeppelin, and Villum, while this pattern changed for Alert and Tiksi
stations (Figure 9). This shows that emissions from Northern Eurasia may extend up to the
American Arctic (Barrie, 1986). The region marked as "other" in Figure 9 stands for all BB
emissions occurring over the North Hemisphere, excluding Northern Eurasia, and shows that
4–42% of the surface concentrations may be due to fires in North America and other BB
sources. In all cases, fires in regions north of 50°N contributed the most, especially to Tiksi
station. Fires in the Asian part of Northern Eurasia contributed to the surface concentrations
of the stations by 13% to 57%, with the maximum at Tiksi station. This was expected since
this station is in the middle of the Northern Eurasian Arctic region and receives a lot of BC
emitted from BB in Siberia. The respective portion for fires occurring in Europe was
estimated to be 1–2% only. Similarly, BC emitted from Siberia contributed 9–43% to the
simulated surface concentrations at the stations (with a maximum in Tiksi station). We
estimated that the total of all vegetation fires in Northern Eurasia contributed around 56 ng m$^{-}$
$^3$, on average, to the Alert station (during spring and summer months), which is close to 89 ng
m$^{-3}$ that Huang et al. (2010) estimated for USSR and European Union, although with different
geographic definitions than those assumed here, and also including anthropogenic emissions.
Gong et al. (2010) and Shindell et al. (2008) estimated that the Northern Eurasian
contribution to Alert varied between 80–90%, while our runs suggested that BB lower
contribution of 29%. Nevertheless, considering our comparison of modelling results with
surface observations for BC from the five Arctic stations, it should be noted that the
calculated contribution constituted an upper bound, especially when considering Asian and
Siberian regions.



## 4.2 Transport efficiency of BC to the Arctic

In this section, we examine the relative roles of different regions to emitting BC that will ultimately be deposited over the Arctic. To do so, we computed the probability of BC emitted from different regions to reach the Arctic. We defined the transport efficiency to the Arctic as the ratio between the mass of BC deposited in the Arctic to the total mass of BC emitted from a given region. These estimates were obtained by masking the same geographical regions as in section 3.3 (anthropogenic sources in the Northern Hemisphere, vegetation fires in Europe, Asia, between 35°N–40°N, 40°N–50°N, 50°N–60°N and above 60°N) and simulating fires that occurred inside the masked areas for the period 2012–2013 (Table 1).

Figure 10 depicts the transport efficiency with which BC from vegetation fires reached the Arctic for the different geographic regions. The results clearly show that anthropogenic BC from large emitting regions in southeastern Asia and other Asian regions were not transported efficiently to the Arctic. The main source contributing to Arctic deposition was BB in Northern Eurasia with a transport efficiency of 10–32% during spring and summer, and 1–6% in autumn and winter. Overall, the transport and subsequent deposition of BC from Asia was more effective than from Europe (12% of the BB emissions from Asia were deposited in the Arctic, whereas only 5% of the European BB emissions reach the Arctic), which was attributed to the fact that European BC tends to remain close to the PBL (Figure 10), whereas the Asian BC mixes up rapidly into the free troposphere (Stohl et al., 2002). Therefore, European BC was much more affected by removal processes since its transport to the Arctic is much less efficient.

In contrast, Siberian BC was deposited very efficiently to the Arctic in summer and autumn similar to fires above 60°N (besides, Siberia covers the Asian part of the 60°N–90°N area). However, it was evident from our results, that in our model the most efficient regional transport of BC to the Arctic occurred in the summer months and was attributed to vegetation fires in Kazakhstan and Mongolia (apart from Siberia, Figure 10). To summarize these results, the highest transport efficiencies in our model occurred in the spring and summer for all Northern Eurasia. This may be a result of (i) extreme fire events, (ii) the relatively weak removal processes occurring in mid- and high-latitudes which favor transport without removal of BC, and (iii) the imposed fixed injection profiles used in these simulations.





## 5  Conclusions

The present study focused on the impact of vegetation fires occurring in Northern Eurasia on BC deposition in the Arctic. For this reason, a three dimensional global transport model (LMDz-OR-INCA) was used to simulate fire events that took place during 2002–2013. Anthropogenic emissions were adopted from MACCity inventory, while BB emissions within Northern Eurasia were from FEI-NE and beyond Northern Eurasia from MACCity's GFEDv3 database.

A total of $3.0 \times 10^6$ km$^2$ was burned during the 12-year period in Northern Eurasia with the majority to be grassland and forest fires. Total global emissions of BC ranged from 8.02 Tg to 9.48 Tg (average: 8.42±0.43 Tg yr$^{-1}$) with the highest ones recorded for the years 2003, 2006, 2008, and 2012. The annual emissions from vegetation fires in Northern Eurasia were estimated to be between 0.45 and 2.19 Tg (average: 0.86±0.51 Tg yr$^{-1}$). Comparing to the MACCity emission inventory, our simulations suggested that 10–17% (average: 8%) more BC was emitted by FEI-NE+MACCity, while FEI-NE biomass burning emissions were 3.5 times higher than GFED3.

The annual mean deposition of BC in the Arctic from vegetation fires in Northern Eurasia was found to be 65±28 kt yr$^{-1}$ for the 12-year period, which represents 45–78% of the BC deposited from all possible sources and origins. The combined run (FEI-NE+MACCity) brought around 55% (1218 versus 675 kt in total for the 12-year period) more BC to deposit over the Arctic environment comparing to the conventional MACCity emission inventory.

Arctic burden showed a strong seasonal variation, which peaks during late winter and early spring in the presence of the Arctic Haze. The peak in winter depicted the latitudinal transport of BC mainly from anthropogenic sources in Europe, whereas the peak in spring and summer clearly stemmed from the fire episodes in Northern Eurasia. The annual mass of BC deposited over the Arctic increased during the most intense fire years (37% in 2009 to 181% in 2012) in comparison to the annual average for the period of 2002–2013.

Fires occurring in the Northern Hemisphere contributed 68% to the simulated deposition of BC in the Arctic, while the rest originated from anthropogenic sources. The majority of the vegetation fires in the Northern Hemisphere was attributed to Northern Eurasian vegetation fires (85%), of which Asia contributed 81% and Europe only 4%. These results were consistent with what other researchers have reported, as Asian BC experienced fast elevation to the free troposphere and hence long–range transport.

The present results were compared to surface observations from five stations (Alert, Barrow, Villum, Tiksi, and Zeppelin) showing relatively good results and in most stations



capturing the trend in surface BC concentrations (a notable deviation was observed at Alert
Alert). We estimated that vegetation fires in Northern Eurasia contributed 14% to 57% to the
surface environment of these stations, mostly affected by fires that took place in Siberia. This
showed the importance of fires occurring over Northern Eurasia in the Arctic BC budget.
However, anthropogenic sources also remain essential contributing 29% to 54% to the surface
of the Arctic stations.
**Acknowledgements**
This study was supported by the US Forest Service, Rocky Mountain Research Station. It was
also granted access to the HPC resources of [CCRT/TGCC/CINES/IDRIS] under the
allocation 2012-t2012012201 made by GENCI (Grand Equipement National de Calcul
Intensif). We would also like to acknowledge the World Data Centre for Aerosol, in which
BC measurements from Arctic stations are hosted (http://ebas.nilu.no). Authors would like to
acknowledge Dan Veber for calibration and instrument maintenance, other technicians,
students and staff of CFS Alert for maintaining the site.



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





1    **TABLE CAPTIONS**

2    **Table 1.** List of simulations from this study to characterize the transport and origin of BC.

| Name | Anthropogenic sources | BB sources | Purpose |
| --- | --- | --- | --- |
| FEI-NE+MACCity | MACCity | FEI-NE | Study of BC transport to the Arctic |
| MACCity | MACCity | MACCity | Comparison with the combined FEI-NE+MACCity run |
| Europe | – | FEI-NE | Study of BC originating from Europe by masking this region |
| Asia | – | FEI-NE | Study of BC originating from Asia by masking this region |
| Siberia | – | FEI-NE | Study of BC originating from Siberia by masking this region |
| Kazakhstan | – | FEI-NE | Study of BC originating from Kazakhstan by masking this region |
| Mongolia | – | FEI-NE | Study of BC originating from Mongolia by masking this region |
| 35°N–40°N (10°W–170°E) | – | FEI-NE | Study of BC originating from latitudes 30°N–40°N in Eurasia by masking this region |
| 40°N–50°N (10°W–170°E) | – | FEI-NE | Study of BC originating from latitudes 40°N–50°N in Eurasia by masking this region |
| 50°N–60°N (10°W–170°E) | | FEI-NE | Study of BC originating from latitudes 50°N–60°N in Eurasia by masking this region |
| 60°N–90°N (10°W–170°E) | – | FEI-NE | Study of BC originating from latitudes >60°N in Eurasia by masking this region |



1 **Table 2.** Comparison between annual BC emissions (Tg) for the period 2002–2013 (FEI-NE+MACCity) and MACCity emissions for the same

2 period. The deposition of BC (ktons or kt) from vegetation fires over the Arctic is also compared to those from the MACCity inventory.

| | 2002 | 2003 | 2004 | 2005 | 2006 | 2007 | 2008 | 2009 | 2010 | 2011 | 2012 | 2013 | Range |
|---|---|---|---|---|---|---|---|---|---|---|---|---|---|
| Anthropogenic sources (Tg) | 5.22 | 5.26 | 5.30 | 5.34 | 5.31 | 5.28 | 5.25 | 5.23 | 5.20 | 5.17 | 5.15 | 5.15 | 5.15–5.34 |
| Anthropogenic sources in Eurasia (Tg) | 2.26 | 2.26 | 2.27 | 2.28 | 2.28 | 2.27 | 2.26 | 2.25 | 2.24 | 2.23 | 2.20 | 2.20 | 2.20–2.28 |
| BB sources (FEI-NE+MACCity) (Tg) | 2.83 | 4.22 | 2.98 | 2.89 | 3.29 | 2.74 | 3.47 | 3.03 | 2.82 | 3.15 | 3.55 | 2.90 | 2.74–4.22 |
| **FEI-NE fires in Eurasia (Tg)** | **0.62** | **2.19** | **0.57** | **0.47** | **0.90** | **0.57** | **1.39** | **0.69** | **0.45** | **0.77** | **1.17** | **0.53** | **0.45–2.19** |
| **FEI-NE+MACCity total (Tg)** | **8.05** | **9.48** | **8.28** | **8.23** | **8.60** | **8.02** | **8.72** | **8.26** | **8.02** | **8.32** | **8.70** | **8.05** | **8.02–9.48** |
| Arctic deposition from fires in Eurasia (kt) | 84 | 98 | 43 | 36 | 88 | 29 | 79 | 42 | 49 | 58 | 120 | 51 | 29–120 |
| Arctic deposition from anthropogenic sources (kt) | 38 | 44 | 38 | 42 | 41 | 36 | 38 | 36 | 33 | 36 | 32 | 28 | 28–44 |
| **Total deposition over the Arctic (kt)** | **122** | **142** | **81** | **78** | **129** | **65** | **117** | **78** | **82** | **94** | **152** | **79** | **65–152** |
| | | | | | | | | | | | | | |
| MACCity anthropogenic sources (Tg) | 5.22 | 5.26 | 5.30 | 5.34 | 5.31 | 5.28 | 5.25 | 5.23 | 5.20 | 5.17 | 5.15 | 5.15 | 5.15–5.34 |
| MACCity anthropogenic sources in Eurasia (Tg) | 2.26 | 2.27 | 2.27 | 2.28 | 2.27 | 2.27 | 2.26 | 2.25 | 2.24 | 2.22 | 2.20 | 2.20 | 2.20–2.28 |
| MACCity BB sources globally (Tg) | 2.51 | 2.47 | 2.55 | 2.57 | 2.34 | 2.64 | 2.04 | 2.62 | 2.62 | 2.62 | 2.62 | 2.62 | 2.04–2.62 |
| **MACCity BB (GFED3) in Eurasia (Tg)** | **0.30** | **0.43** | **0.13** | **0.13** | **0.21** | **0.14** | **0.25** | **0.24** | **0.24** | **0.24** | **0.24** | **0.24** | **0.13–0.43** |
| **MACCity total (Tg)** | **7.73** | **7.73** | **7.85** | **7.91** | **7.65** | **7.92** | **7.29** | **7.85** | **7.82** | **7.79** | **7.77** | **7.76** | **7.29–7.92** |
| Wet deposition over the Arctic (kt) | 64 | 65 | 50 | 49 | 54 | 40 | 52 | 52 | 51 | 51 | 50 | 42 | 42–65 |
| Dry deposition over the Arctic (kt) | 5 | 6 | 6 | 6 | 4 | 3 | 3 | 5 | 5 | 5 | 5 | 4 | 3–6 |
| **Total deposition over the Arctic (kt)** | **69** | **71** | **56** | **55** | **58** | **43** | **55** | **57** | **56** | **56** | **55** | **46** | **43–71** |





**FIGURES**

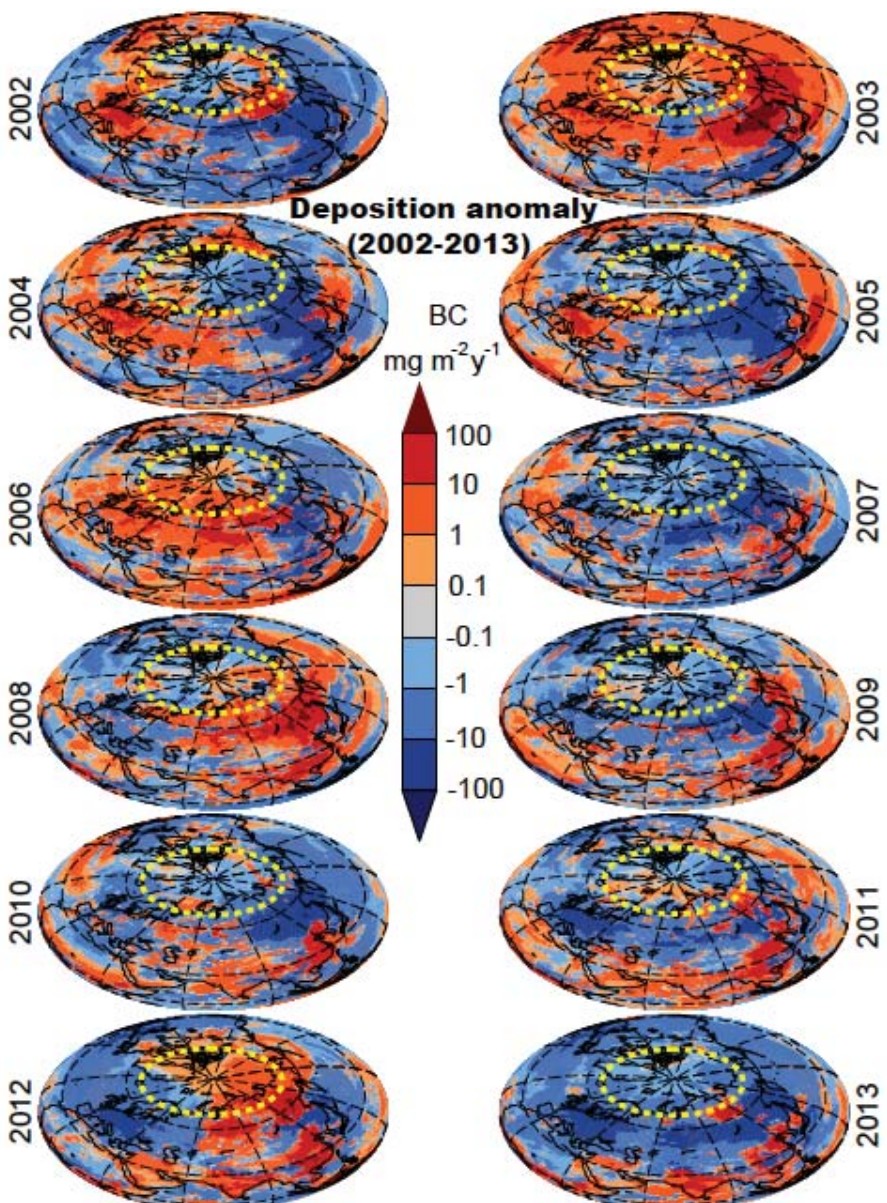

**Figure 1.** Deposition anomalies of BC (mg m$^{-2}$ y$^{-1}$) in the North Hemisphere for the period
2002–2013 from our combined simulation (FEI-NE+MACCity) in Northern Eurasia. The
dashed yellow line represents the border of the Arctic (~67°N). Note that the most intense
years were estimated to be 2003, 2006, 2008, and 2012.





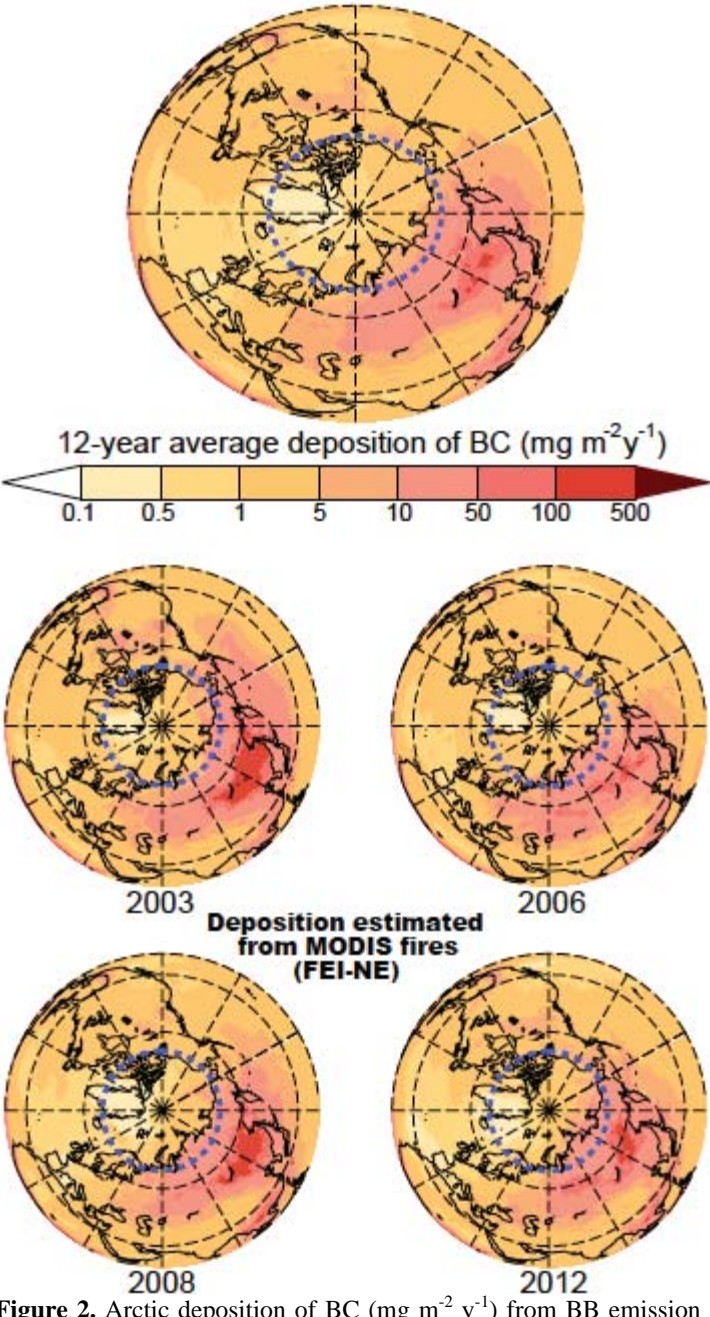

**Figure 2.** Arctic deposition of BC (mg m$^{-2}$ y$^{-1}$) from BB emission according to the FEI-NE
inventory. The upper panel depicts the 12-year average deposition, while the lower 4 panels
show the most intense fire years (2003, 2006, 2008, and 2012). The dashed blue line
represents the border of the Arctic (~67°N).





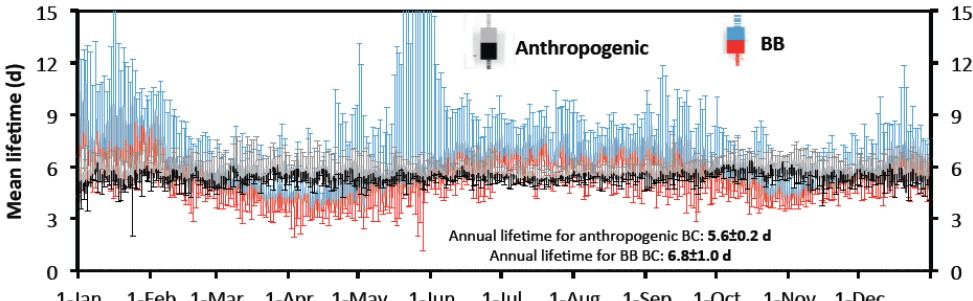

**Figure 3.** Annual mean lifetime of the global anthropogenic and BB BC from our combined
FEI-NE+MACCity simulation. The results are presented as Box & Whisker plots of daily
lifetimes of BC (both for anthropogenic and BB) for the period 2002–2013. The plot shows
the minimum value, the 25[th] percentile, which holds 25% of the values at or below it. The
median is the 50[th] percentile, the third quartile is the 75[th] percentile and the maximum is the
100[th] percentile (i.e., 100% of the values are at or below it).

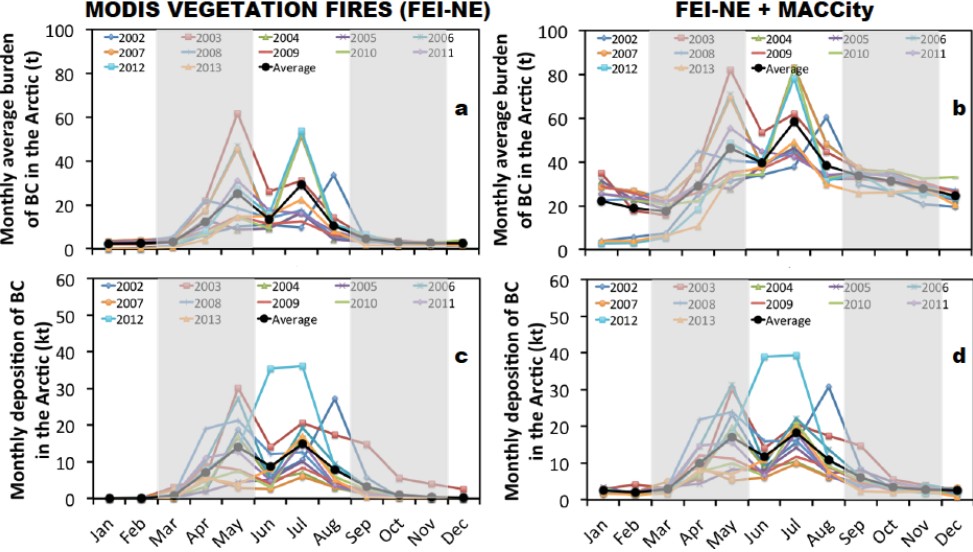

**Figure 4.** Monthly atmospheric burden of BC (t) in the Arctic with each year between 2002
and 2013 represented by a different colored line: **(a)** from vegetation fires (FEI-NE) only and
**(b)** from all BC emissions (FEI-NE+MACCity). Panel c and d shows the same, but for the
Arctic deposition of BC (kt).



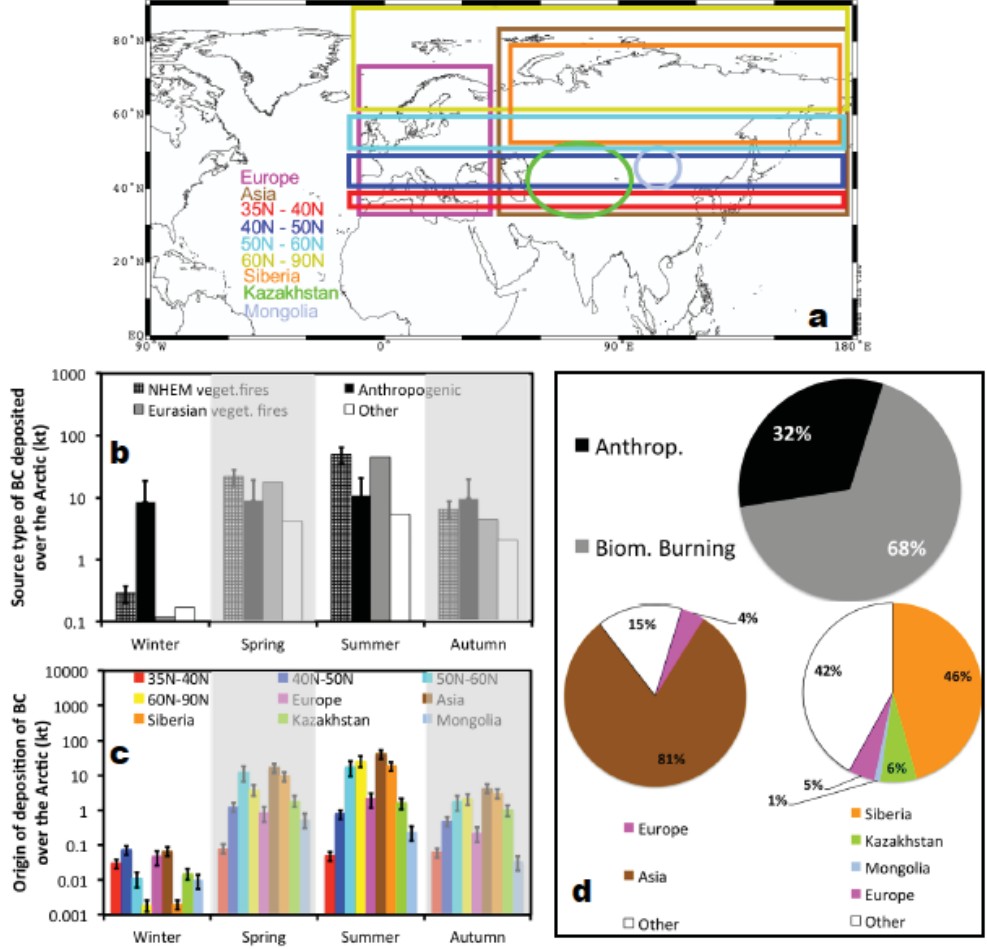

**Figure 5. (a)** The nine geographic regions of BC emission in Northern Eurasia defined for

this study. **(b)** Seasonal values of BC emitted from all fires in Northern Hemisphere (NHEM),

anthropogenic sources (black) and Northern Eurasian vegetation fires (grey) deposited in the

Arctic for the period of 2002–2013. **(c)** Contribution of several geographic regions to the

Arctic BC deposition. Colors are used according to the ones used in panel a. Red stands for

regions within 35°N–40°N, dark blue for 40°N–50°N, turquoise for 50°N–60°N, yellow for

regions located above 60°N (60°N–90°N), magenta for Europe, brown for Asia, orange

denotes Siberia, green for Kazakhstan, and light blue for Mongolia. **(d)** Pie-charts showing

the origin of BC deposition in the Arctic from vegetation fires in Northern Eurasia.



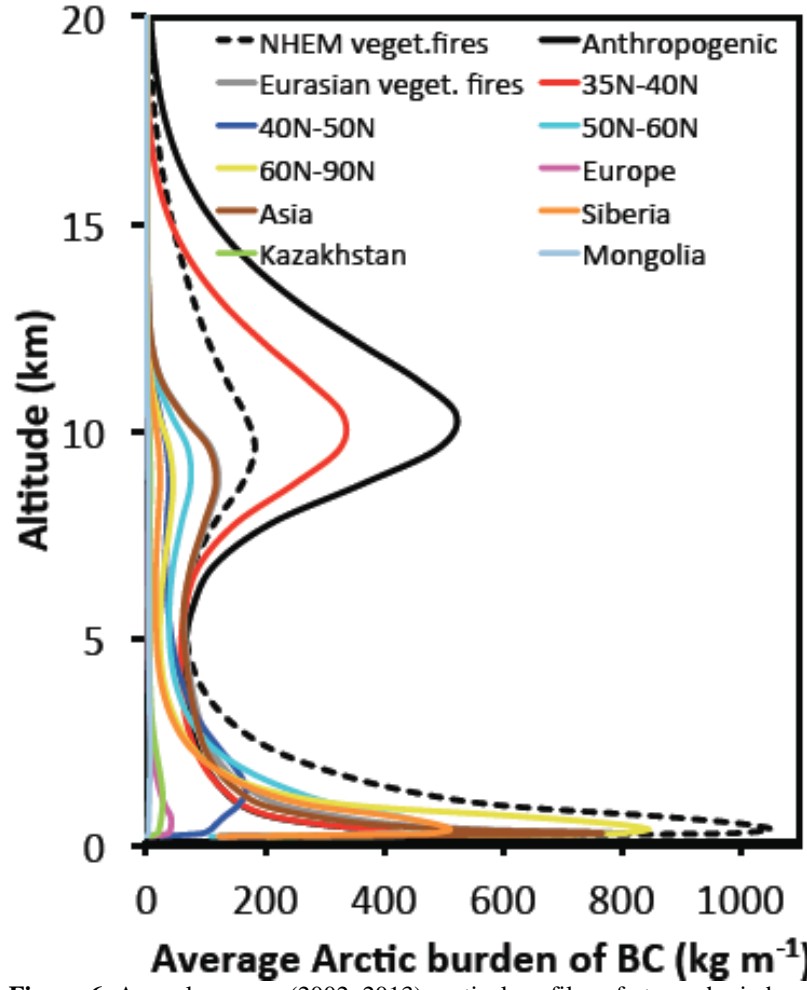

2      **Figure 6.** Annual average (2002–2013) vertical profiles of atmospheric burden of BC (kg m⁻

3      ¹) in the Arctic originating from different regions.





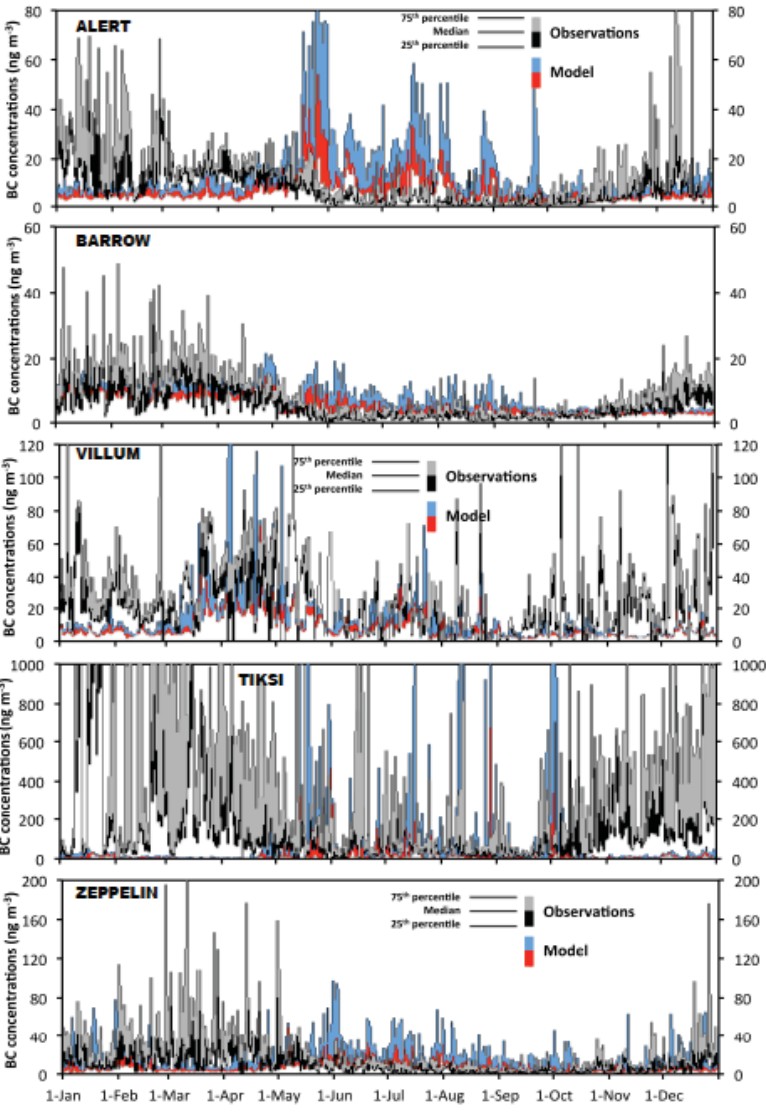

**Figure 7.** Modeled versus measured surface concentrations of BC (ng m$^{-3}$) for the FEI-NE+MACCity simulation in the Arctic stations Alert, Barrow, Villum, Tiksi, and Zeppelin. Due to the high variability of the surface concentrations, the results are presented as Box & Whisker plots of all the modeled and measured surface daily concentrations of BC for the period 2002–2013. The plots show the minimum value, the 25$^{th}$ percentile, which holds 25% of the values at or below it. The median is the 50$^{th}$ percentile, the third quartile is the 75$^{th}$ percentile and the maximum is the 100$^{th}$ percentile (i.e., 100% of the values are at or below it).



**Figure 8.** Difference in BC atmospheric burden (mg m$^{-3}$) between our simulation that combines emission inventories (FEI-NE+MACCity) and MACCity. The dashed yellow line represents the limit of the Arctic (~67°N). The BC burden was estimated by averaging all the vertical layers over 365 days for each year from 2002 to 2013.



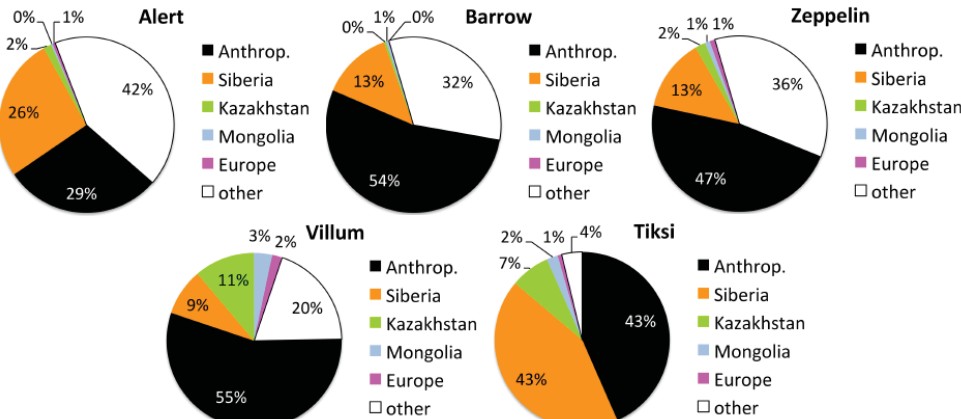

**Figure 9.** Multi-year average (2002–2013) contribution of global anthropogenic and BB BC from the respective geographic regions to surface concentrations at the Arctic stations (Alert, Barrow, Zeppelin, Villum, and Tiksi). "Other" stands for other locations where BB sources were not accounted for.

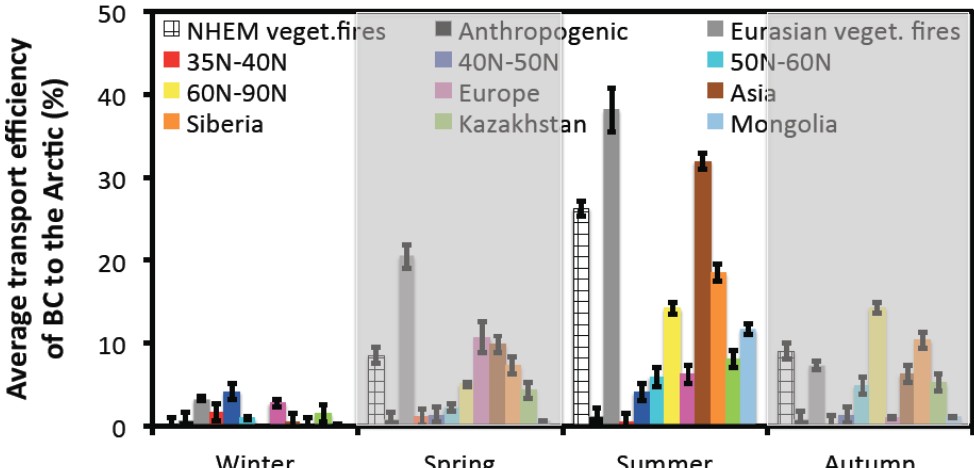

**Figure 10.** Relative transport efficiency of BC from vegetation fires from different geographic regions across the Arctic. The same colors were used as in **Figure 5**.