# Peer review of "Wildfires in Northern Eurasia affect the budget of black"

_Atmospheric Chemistry and Physics, 2015_

## Short Comment (SC1) · 16 Mar 2016

Dear authors,

Thank you for your contribution, I enjoyed reading your paper. However, one thing puzzles me and I would be happy if you could elaborate on that. In your paper, you come to the following conclusion: "Northern Eurasian vegetation fires (FEI-NE) contributed 85 % (79–91 %) to the BC deposited over the Arctic from all BB sources in the Northern Hemisphere." When I understood the paper correctly, this result is based on the run where you used different fire emission inventories for Northern Eurasia and the rest of the world (FEI-NE+MACCity). Is this statement not dangerous since you showed that the FEI-NE emissions are considerably higher than the MACCity emissions over the

same region (Northern Eurasia)? Your conclusion could partly be due to the different emission inventories, and you might underestimate the influence of fire emissions from North America. I am looking forward to your answer and some further discussions.

---

## Short Comment (SC2) · 16 Mar 2016

Dear Dr. Gilgen, Thank you very much for pointing this problematic statement. I agree that we may underestimate the impact of BB in N. America that, in our study, originates from GFED. We will certainly re-formulate these sentences, as it seems we insinuate that the BB emissions estimated from FEI-NE are 100% valid/real. However, some recent top-down estimations of BC emissions suggest that the majority of underestimated emission sources is rather located in Eurasia (some originates from N. America, as well). These are BB sources, which in turn are underestimated in GFED, whereas some others are anthropogenic (mainly from gas-flaring as stated in Stohl et al., 2013) that have been poorly addressed.

---

## Referee Comment (RC1) · Anonymous Referee #1 · 24 Mar 2016

Review: Wildfires in Northern Eurasia affect the budget of black carbon in the Arctic. A 12-year retrospective synopsis (2002–2013).

The manuscript introduces a modeling study applying BC emission inventories in an atmospheric chemistry transport model to analyse the deposition of BC in the Artic stemming from Northern Eurasia. Several sensitivity simulations were performed to disentangle the contribution of different regions within the Northern Hemisphere. This is an interesting and relevant topic and the methods used in the study sound valid. The manuscript, however, would benefit from a better structuring of the results. In general, I do recommend publication, but suggest a number of changes.

General comments:

1. The study applies the new biomass burning emission dataset by Hao et al. for Northern Eurasia (NE). How this emission dataset has been derived has to be discussed more in detail in this paper. Particularly, the differences to the GFEDv3 emissions have to be outlined as these are applied in this study as well.

2. Naming of the experiments: MACCity-FEI-NE and MACCity simulation do differ only in the representation of biomass burning in NE. One uses Hao et al., the other one GFEDv3. I'd suggest that the simulations are renamed to more explicitly reflect this differences (e.g. FEI-NE and GFEDv3).

3. The difference between the simulations MACCity-FEI-NE and MACCity have to be discussed more in detail. Here the manuscript would benefit from a comparison of the MACCity simulation with the observations and not only the comparison MACCity-FEI-NE and observations.

4. At the same time, the discussion of the region specific simulation should only refer to the MACCity-FEI-NE simulation and it has to made clear throughout the manuscript that the conclusion are based on the MACCity-FEI-NE settings.

The abstract is way too long and should be shortened.

Page1/Line12: estimated is not the right term here – used?

Page1/Line 14: is this area based on FEI-NE or GFEDv3? Is the global number based on GFEDv3?

Page1/Line16: 70% is this for the FEI-NE or GFEDv3?

Page1/Line19: " . . . was twice as much as when using MACCity " , i.e. twice as much as when excluding biomass burning emissions? Maybe here and in the following it would be easier for the reader to follow when you refer to anthropogenic emissions in more general and not specifically to the MACCity inventory. You mentioned in the beginning that anthropogenic emissions are used from MACCity.

Page1/Line23: As mentioned in another comments here it must be made clear what emission inventories these numbers refer to. All regions based on GFEDv3, or northern Eurasia set to FEI-NE? Best is both scenarios are mentioned.

Introduction:

Page5/Line9: this argument is already given in the paragraph above. Please combine The introduction should also briefly discuss the emission inventories available for biomass burning in NE. These make up a substantial part of the paper and the conclusions.

Page7/Line26: isn't it 2005 and not 2000?

Page8/Line 25: And what injection height is used outside NE?

2.3 BC emissions . . ..

I do find the naming convention not that intuitive. Why don't you use FEI-NE and GFEDv3 for Biomass burning and MACCity for the anthropogenic. That GFEDv3 is part of MACCity is not that obvious and a bit hidden in the manuscript.

Page 11/Line2: from the FEI-NE+MACCity and the MACCity simulation" →used/applied in the . . . and . . . simulation.

Page 11/Line10: Shouldn't there be a difference between FEI-NE and MACCity for the global number?

Page 11/Line12: Tg – Tg/year here and in the following.

Page 11/Line17: Why do you reference Bond et al., Isn't this number based on your study?

Table2:

- that the anthropogenic sources are listed twice is confusing. Also the numbers should be identical but this is not the case for some of the years.

- artic deposition from NE fires and artic deposition from anthropogenic sources do add to the total artic deposition, this can not be correct.

Page11/ Line 20: which four years do you refer to 2006, 2003 and ? and do you refer to global or NE values?

Page 11/Line13: "This indicates that during these years the largest amounts of BC were deposited over Arctic regions as a result of large fire events in Siberia, Western Russia, and Kazakhstan. " – I don't see here how you reached this conclusion.

Page 13: The deposition rates for the artic results form all sources have to be discussed for both emission inventories. In addition, a simulation evaluation the contribution of NE fires to the Artic deposition based on GFEDv3 would be valuable for comparison.

Page13/Line 29: but fire detection is not directly related to fire emissions.

Page14/Line 18: I do not understand how the anthropogenic fraction is derived and what it exactly refers to (all anthropogenic, anthropogenic from NE?). Please clarify here and in Figure 5d.

Page16/Line1: Figure 7 compares the surface concentrations not Figure 8

Page16/Line7: Figure 7 compares the simulated versus observed daily surface concentrations by a Box and Whisker – what is blue and red for the model results? Also, it would be interesting to compare to the observations also the MACCity simulation. Does the different representation of fire in NE in the FEI-NE simulation actually improve the model results?

Page16/Line 20: How do you distinguish in the plot between anthropogenic and BB sources?

Page17/ Line 25: Here you have to be more specific. Differences arise mainly from the fact that the BC emissions are lower in NE (a region you identified as being important for artic BC deposition) and not so much from the fact that global depositions are

reduced.

Page17/ Line28: Why do you derive the importance of NE fires here from the difference of the MACCity-FEI-NE and MACCity simulation for atmospheric burden, etc. and not from the simulation were you excluded fires in NE as in the previous paragraph? More interesting would be a comparison of MACCity with observations.

Page 18/Line1: 'We also analyzed the influence of all anthropogenic and BB emissions from the regions (defined in Table 1) to the average surface concentration of the Arctic stations (Figure 9). ' – but the anthropogenic sources are only assessed globally and not by region.

Page18/Line 14: Region 'other' . The explanation here reads different from the figure caption.

Page 18/Line19: " . . . while our runs suggested that BB lower contribution of 29%" – please correct

Page19/Line 12: but you didn't explicitly differentiate for different anthropogenic source regions – or did I miss something?

Page 20/Line8: 3.0 *10ˆ6 or 250.000 as stated in the abstract?

Page20/Line14: 3.5 times higher in NE or globally?

---

## Referee Comment (RC2) · Anonymous Referee #2 · 7 Apr 2016

The authors present a set of multi year simulations of northern hemisphere BC transport and deposition, based on two emission inventories. They focus on the Arctic, and estimate the contribution from Northern Eurasian Wildfires. The results in the paper are relevant to the ongoing discussion on the atmospheric, environmental and climate effects of black carbon. Their methods are sound and relatively standard, and the presentation acceptable - although I have some comments and suggestions. I recommend that the paper be published in ACP after some revisions, mainly regarding the clarity of some of the arguments presented.

Major comments

- While the authors present results from one model (LMDZ-OR-INCA), they also compare their results to other studies. To fully make this comparison, I recommend adding a brief discussion on how their model has performed relative to others in recent multi-model comparisons, notably AeroCom Phase II (Myhre et al. 2013, ACP).

- Throughout, I also miss some simple sensitivity studies for the key or updated parameters of model, and some discussion of how robust the authors expect that their results are. E.g. on page 6 they state that " A comparison made with inert tracers indicated an enhanced vertical transport as the horizontal resolution of the model was increased from 144×142 grid-points to 280×192." Does this have any bearing on the results here? If so, what is the impact of this enhancement on the burdens and vertical profiles presented later? And on page 14 they state that " the annual mean lifetime of anthropogenic BC particles from BB was longer (6.8 d) than for BC from combustion (5.6 d)". However the uncertainties given above indicate that these values are consistent within errors. (What are the errors? One sigma? I cannot find this specified.) Figure 3 further indicates that there's little significant difference between the two estimates.

- On the lifetime calculation and Figure 3: Since the authors use a steady state definition (which I agree is reasonable), and the lifetime is on the order of a week, is it really meaningful to show daily lifetime values?

- I would recommend a thorough reworking of the figures. While they are well thought out and have the right content, they are often very hard to interpret. For the map plots, e.g. Figure 1, the resolution is low (perhaps just a feature of ACP processing), and the continent lines virtually invisible. A polar projection like Figure 2 is more readable, even if it skews the outer edge. In Figure 3 the whiskers come out OK, but the box mentioned in the caption is invisible. Same for Figure 7. Figures 5 and 10 have grey boxes overlaying the figure content (again possibly a processing issue, but please check).

Minor comments

- The abstract is quite lengthy. I would recommend shortening it, as brief abstracts

greatly increase the readability of papers. The main results are anyway repeated in the Conclusions.

- Page 16: Two references to Figure 8 should be Figure 7.

- Figure 6: Is there any interannual variability in the shape of these vertical profiles? This is an interesting observable quantity for estimating transport. Your years cover the HIPPO flight times (Schwarz et al. 2013, GRL). Have you considered a comparison here? Even without this, many studies use HIPPO, and it would be interesting to know how stable the conditions seen by those flights were likely to have been.

- Figure 8: How was the vertical level averaging done? The caption states "averaging all the vertical layers". The standard definition of atmospheric burden is the sum of the abundances in each layer (i.e. concentration times the height of the layer). If this is not what is shown here, another word than burden should be used.

---

## Author Comment (AC1) · 5 May 2016

Review: Wildfires in Northern Eurasia affect the budget of black carbon in the Arctic. A 12-year retrospective synopsis (2002–2013).

The manuscript introduces a modeling study applying BC emission inventories in an atmospheric chemistry transport model to analyse the deposition of BC in the Artic stemming from Northern Eurasia. Several sensitivity simulations were performed to disentangle the contribution of different regions within the Northern Hemisphere. This is an interesting and relevant topic and the methods used in the study sound valid. The manuscript, however, would benefit from a better structuring of the results. In general, I do recommend publication, but suggest a number of changes.

General comments:

1. The study applies the new biomass burning emission dataset by Hao et al. for Northern Eurasia (NE). How this emission dataset has been derived has to be discussed more in detail in this paper. Particularly, the differences to the GFEDv3 emissions have to be outlined as these are applied in this study as well.

Response: All these aspects and comparisons are discussed in our companion paper (Hao, W. M., et al., Geosci. Model Dev. Discuss., doi:10.5194/gmd-2016-89, in review, 2016), where a great analysis is given. Discussing the same here, while the GMD paper is in revision in an open-access journal would be rather inappropriate. However, if the reviewer is still not happy, we could add some more details in a next step, although this would degrade Hao et al. publication in GMD.

2. Naming of the experiments: MACCity-FEI-NE and MACCity simulation do differ only in the representation of biomass burning in NE. One uses Hao et al., the other one GFEDv3. I'd suggest that the simulations are renamed to more explicitly reflect this differences (e.g. FEI-NE and GFEDv3).

Response: We would partially agree with the reviewer and he/she seems to understand very well the notion of the present study.

However, we do not agree with the new names that he/she suggests. We believe that naming MACCity-FEI-NE and MACCity simulation as FEI-NE and GFEDv3 is not accurate at all. GFEDv3 is a global dataset and FEINE refers to an approach applied over a certain region (Northern Eurasia). In our opinion, this would confuse the reader very much making him believe that FEINE is a global dataset, which unfortunately is not the case yet. Besides, we explain very much in detail what our runs refer to in Table of the manuscript and in the Abstract and Methodology sections.

3. The difference between the simulations MACCity-FEI-NE and MACCity have to be discussed more in detail. Here the manuscript would benefit from a comparison of the MACCity simulation with the observations and not only the comparison MACCity-FEI-NE and observations.

Response: Corrected. We have put in Fig.7 surface concentrations from MACCity itself and we show them together with the respective observations and FEI-NE+MACCity concentrations. We further discuss which of the datasets perform better in Discussions.

4. At the same time, the discussion of the region specific simulation should only refer to the MACCity-FEI-NE simulation and it has to made clear throughout the manuscript that the conclusion are based on the MACCity-FEI-NE settings.

Response: Corrected (beginning of 4.2 section). In addition, it is shown in Table 1. The abstract is way too long and should be shortened. Page1/Line12: estimated is not the right term here – used?

Response: Corrected to "adopted from".

Page1/Line 14: is this area based on FEI-NE or GFEDv3? Is the global number based on GFEDv3?

Response: Corrected. It is now clarified inside parentheses.

Page1/Line16: 70% is this for the FEI-NE or GFEDv3?

Response: Corrected. It is now clarified inside parentheses.

Page1/Line19: " . . . was twice as much as when using MACCity ", i.e. twice as much as when excluding biomass burning emissions? Maybe here and in the following it would be easier for the reader to follow when you refer to anthropogenic emissions in more general and not specifically to the MACCity inventory. You mentioned in the beginning that anthropogenic emissions are used from MACCity.

Response: It is now corrected, in order to be clearer.

Page1/Line23: As mentioned in another comments here it must be made clear what emission inventories these numbers refer to. All regions based on GFEDv3, or northern Eurasia set to FEI-NE? Best is both scenarios are mentioned.

Response: Numbers have been added in the abstract, as the reviewer suggested.

Introduction:

Page5/Line9: this argument is already given in the paragraph above. Please combine The introduction should also briefly discuss the emission inventories available for biomass burning in NE. These make up a substantial part of the paper and the conclusions.

Response: Corrected as the reviewer suggested. As regards to the emissions from BB in Nor. Eurasia, all details have been included in the Methodology.

Page7/Line26: isn't it 2005 and not 2000?

Response: In Lamarque et al. (2010) it is clearly mentioned that the dataset exists until 2000. Then ACCMIP was extended until 2100 (http://www.geosci-model-dev.net/6/179/2013/gmd-6-179-2013.html)

Page8/Line 25: And what injection height is used outside NE? 2.3 BC emissions . . ..

Response: Corrected. Now, we clearly write in the manuscript that a similar injection profile for biomass burning outside Nor. Eurasia was used with emissions occurring up to 1000 m. I do find the naming convention not that intuitive. Why don't you use FEI-NE and GFEDv3 for Biomass burning and MACCity for the anthropogenic. That GFEDv3 is part of MACCity is not that obvious and a bit hidden in the manuscript.

Response: As I mentioned in a previous comment the names that the reviewer suggested are not accurate. GFED3 corresponds to a global dataset, which is not the case for FEI-NE. We explain in detail what MACCity and FEI-NE+MACCity correspond to in the methodology and we show what input each of the simulations used in Table 1.

Page 11/Line2: from the FEI-NE+MACCity and the MACCity simulation"
→used/applied in the . . . and . . . simulation.

Response: The sentence has slightly changed.

Page 11/Line10: Shouldn't there be a difference between FEI-NE and MACCity for the global number?

Response: Corrected. It is FEINE+MACCity instead of FEINE and MACCity.

Page 11/Line12: Tg – Tg/year here and in the following.

Response: Corrected everywhere in the manuscript according to reviewer's suggestion.

Page 11/Line17: Why do you reference Bond et al., Isn't this number based on your study?

Response: We acknowledge reviewer for this correct comment. This reference does not match, because the results are based on our study.

Table2:

- that the anthropogenic sources are listed twice is confusing. Also the numbers should be identical but this is not the case for some of the years.

Response: Anthropogenic source are listed twice, in order to show that they have been adopted from the same dataset in Eurasia (and elsewhere). As the Table looks now, it has 2 sets of information, first for FEINE+MACCity and for MACCity for comparison. Each set has basically 6 components, (a) Anthropogenic sources (Tg), (b) Anthropogenic sources in Eurasia (Tg), (c) BB sources (FEI-NE+MACCity) (Tg), (d) FEI-NE fires in Eurasia (Tg), (e) FEI-NE+MACCity total (Tg), and (f) Total deposition over the Arctic (kt). We believe that if we exclude (b) Anthropogenic sources in Eurasia (Tg)

from one of the datasets, it will be confusing for the reader. However, we agree with the reviewer about the slight difference in the values. This can be confusing and we have corrected it.

- Artic deposition from NE fires and artic deposition from anthropogenic sources do add to the total artic deposition, this cannot be correct.

Response: We appreciate for this comment. We have corrected this part. The label was not correct. We stated "Arctic deposition from anthropogenic sources", but it is actually "Arctic deposition from all sources outside Eurasia". We have corrected this part everywhere in the text now.

Page11/ Line 20: which four years do you refer to 2006, 2003 and ? and do you refer to global or NE values?

Response: Corrected! FEI-NE emissions refer always to biomass burning in Northern Eurasia. It is clearly stated in the Methodology.

Page 11/Line13: "This indicates that during these years the largest amounts of BC were deposited over Arctic regions as a result of large fire events in Siberia, Western Russia, and Kazakhstan. " – I don't see here how you reached this conclusion.

Response: We have reformulated this sentence, also pointing the reader to Fig.S2 with the emission anomalies of BC.

Page 13: The deposition rates for the artic results form all sources have to be discussed for both emission inventories. In addition, a simulation evaluation the contribution of NE fires to the Artic deposition based on GFEDv3 would be valuable for comparison.

Response: The goal of the present paper is to identify the role of wildfires over Eurasia using a new approach on the budget of BC in the Arctic and NOT to discuss the differences of FEINE with GFED. This is discussed in detail in our companion paper published by GMD Discussions (http://www.geosci-model-dev.net/6/179/2013/gmd-6-179-2013.html). Nevertheless, we have added more lines on this direction, like the reviewer suggested.

Page13/Line 29: but fire detection is not directly related to fire emissions.

Response: According to our companion paper "The burned area mapping method, which was originally developed for the western United States (Urbanski et al., 2011), has two steps. First, a burn scar algorithm is applied to pixels of the surface reflectance product to identify potential burn scars. Then, the potential burn scars are screened for false detections using a contextual filter that eliminates pixels not proximate with recent active fire detections. For mapping burned areas in Northern Eurasia, the burn scar algorithm was unchanged; however, the contextual filter was modified. In this study, potential burn scars not within 5 km and 10 days of active fire detection were classified as false detections and were eliminated." We have modified "fire detections" to "burn scars" to be more consistent to our companion paper.

Page14/Line 18: I do not understand how the anthropogenic fraction is derived and what it exactly refers to (all anthropogenic, anthropogenic from NE?). Please clarify here and in Figure 5d.

Response: This is answered in the abstract (page 2 – line 12). We account for anthropogenic (MACCity) and biomass burning sources (FEINE in Eurasia – GFED3 outside Eurasia). However, we have corrected this part in Figure 5d as the reviewer suggested.

Page16/Line1: Figure 7 compares the surface concentrations not Figure 8

Response: Corrected!!

Page16/Line7: Figure 7 compares the simulated versus observed daily surface concentrations by a Box and Whisker – what is blue and red for the model results? Also, it would be interesting to compare to the observations also the MACCity simulation. Does the different representation of fire in NE in the FEI-NE simulation actually improve the model results?

Response: Corrected. We have added surface concentrations of BC from MACCity

in the Figure 7 as the reviewer suggested. Regarding if this new approach for the BB emissions improves the results, it appears that it does in some of the stations. However, it appears that anthropogenic BC is misleading in some of the stations.

Page16/Line 20: How do you distinguish in the plot between anthropogenic and BB sources?

Response: Of course, we do not distinguish between anthropogenic and bb in Figure 8. The statements of this paragraph can be easily obtained if one observes emissions of BC. This is done in Figure S2. We now point the reader to this Figure, as well. We appreciate reviewer's help.

Page17/ Line 25: Here you have to be more specific. Differences arise mainly from the fact that the BC emissions are lower in NE (a region you identified as being important for artic BC deposition) and not so much from the fact that global depositions are reduced.

Response: We have slightly changed this part. For your consideration, we do not imply that emissions are lower in NE. The opposite though; we have shown throughout the manuscript, but also in Table 2 that the emissions from FEI-NE are larger in NE (comparing to GFED3). There should not be any misunderstanding now. Page17/ Line28: Why do you derive the importance of NE fires here from the difference of the MACCity-FEI-NE and MACCity simulation for atmospheric burden, etc. and not from the simulation were you excluded fires in NE as in the previous paragraph? More interesting would be a comparison of MACCity with observations.

Response: Corrected. 2-3 sentences about comparison of MACCity with observations have been added according to the reviewer's suggestion.

Page 18/Line1: 'We also analyzed the influence of all anthropogenic and BB emissions from the regions (defined in Table 1) to the average surface concentration of the Arctic stations (Figure 9). ' – but the anthropogenic sources are only assessed globally and not by region.

Response: Corrected. We have clarified the sentence now. As reviewer noticed anthropogenic sources are assessed globally. The contribution to surface concentrations from anthropogenic BC is not country-specific, but rather global. Only BC from FEI-NE is country-specific in this chapter. This is consistent with Table 1.

Page18/Line 14: Region 'other' . The explanation here reads different from the figure caption.

Response: Corrected!

Page 18/Line19: " . . . while our runs suggested that BB lower contribution of 29%" – please correct

Response: Corrected!

Page19/Line 12: but you didn't explicitly differentiate for different anthropogenic source regions – or did I miss something?

Response: As the reviewer pointed out, we do not mask anthropogenic sources in each of the geopolitical regions. The contribution of anthropogenic BC to surface concentrations is calculated globally and it is not country-specific. Furthermore, we made numerous sensitivity runs allowing biomass-burning emissions from a certain area (each time). Hence, one run with only biomass burning emissions from Mongolia (and nowhere else), one from Siberia, one from Asia, one from Europe and so on... This allowed us to estimate how much BC was deposited over the Arctic from emissions occurring over a certain country, continent, etc...

Page 20/Line8: 3.0 *10Ë Ę6 or 250.000 as stated in the abstract? Page20/Line14: 3.5 times higher in NE or globally?

Response: Both burned areas are correct. In the Abstract, we give an annual average burned area, which is 250,000 km2 per yr, while in Conclusions we give a total burned area (for the 12-y period), which is 3,000,000 km2

Please also note the supplement to this comment:
http://www.atmos-chem-phys-discuss.net/acp-2015-994/acp-2015-994-AC1-supplement.pdf

[Figure]

**Supplement:**

[revised manuscript text omitted]

Nikolaos Evangeliou 4/5/2016 23:38

Nikolaos Evangeliou 4/5/2016 23:39

Table 2

---

## Author Comment (AC3) · 5 May 2016

The authors present a set of multi year simulations of northern hemisphere BC transport and deposition, based on two emission inventories. They focus on the Arctic, and estimate the contribution from Northern Eurasian Wildfires. The results in the paper are relevant to the ongoing discussion on the atmospheric, environmental and climate effects of black carbon. Their methods are sound and relatively standard, and the presentation acceptable - although I have some comments and suggestions. I

recommend that the paper be published in ACP after some revisions, mainly regarding the clarity of some of the arguments presented.

Major comments

- While the authors present results from one model (LMDZ-OR-INCA), they also compare their results to other studies. To fully make this comparison, I recommend adding a brief discussion on how their model has performed relative to others in recent multimodel comparisons, notably AeroCom Phase II (Myhre et al. 2013, ACP).

Response: Corrected!!! A brief discussion has been added to Methodology about how LMDZORINCA performed in this intercomparison exercise.

- Throughout, I also miss some simple sensitivity studies for the key or updated parameters of model, and some discussion of how robust the authors expect that their results are. E.g. on page 6 they state that " A comparison made with inert tracers indicated an enhanced vertical transport as the horizontal resolution of the model was increased from $144 \times 142$ grid-points to $280 \times 192$." Does this have any bearing on the results here? If so, what is the impact of this enhancement on the burdens and vertical profiles presented later? And on page 14 they state that " the annual mean lifetime of anthropogenic BC particles from BB was longer (6.8 d) than for BC from combustion (5.6 d)". However the uncertainties given above indicate that these values are consistent within errors. (What are the errors? One sigma? I cannot find this specified.) Figure 3 further indicates that there's little significant difference between the two estimates.

Response: Corrected!! We have added a paragraph discussing about the robustness of our model in end of Page – beginning of Page 6. The comparison with inert gases and the comparison of the vertical transport between the 2 model resolutions has a significant impact in out results, because it implies that it may be an underestimation of surface concentrations due to the enhanced vertical transport. As regards to the second part of the comment, we first need to clarify to the reviewer that 5.6 $\pm$0.2

d and 6.8 ±1.0 d are annual global average lifetimes for anthropogenic and bb BC plus/minus the standard deviation of the dataset (N=365). It is now specified in the caption of Figure 3.

- On the lifetime calculation and Figure 3: Since the authors use a steady state definition (which I agree is reasonable), and the lifetime is on the order of a week, is it really meaningful to show daily lifetime values?

Response: The motivation to show daily lifetimes was just to examine the variation of lifetime with respect to the different origin of BC (anthropogenic, biomass burning). We have made a better Figure 3 now showing only timeseries of minimum, maximum and average lifetimes, which we think that it would be more consistent to how other researchers present such kind of results (see for instance Fig. 3 in Croft et al., 2014).

- I would recommend a thorough reworking of the figures. While they are well thought out and have the right content, they are often very hard to interpret. For the map plots, e.g. Figure 1, the resolution is low (perhaps just a feature of ACP processing), and the continent lines virtually invisible. A polar projection like Figure 2 is more readable, even if it skews the outer edge. In Figure 3 the whiskers come out OK, but the box mentioned in the caption is invisible. Same for Figure 7. Figures 5 and 10 have grey boxes overlaying the figure content (again possibly a processing issue, but please check).

Response: Corrected! As regards to the poor resolution, it is subject to the initial submission process required by the ACPD journal. Of course, the same figures exist in a higher resolution, as well, and they will be submitted when required. We agree with the reviewer about Figures 3 and 7 and we have recreated them. We do not show Box & Whisker plots anymore, but only average lines shaded from minimum to maximum, which is more or less the same.

Minor comments

- The abstract is quite lengthy. I would recommend shortening it, as brief abstracts

greatly increase the readability of papers. The main results are anyway repeated in the Conclusions.

Response: Corrected. We agree with the reviewer and we have corrected according to his suggestion!! We have shortened the abstract to 1 page removing all the unnecessary information.

- Page 16: Two references to Figure 8 should be Figure 7.

Response: Corrected!!

- Figure 6: Is there any interannual variability in the shape of these vertical profiles? This is an interesting observable quantity for estimating transport. Your years cover the HIPPO flight times (Schwarz et al. 2013, GRL). Have you considered a comparison here? Even without this, many studies use HIPPO, and it would be interesting to know how stable the conditions seen by those flights were likely to have been.

Response: We have not compared our profiles with HIPPO, mainly because the HIPPO campaigns were located in a different location than the one of our interest (Eurasia). A fast comparison is shown in Fig. 1 (plot is from HIPPO, Schwarz et al., 2010) and Fig.2 (our model)

The MMR values of the x-axis were far away the HIPPO ones.

In the plot referring our model, it is obvious what was mentioned in a previous comment that the reviewer made about what the impact of the statement "A comparison made with inert tracers indicated an enhanced vertical transport as the horizontal resolution of the model was increased from 144×142 grid-points to 280×192" is in our results. You may see now that comparing to HIPPO, a larger mass has been transported vertically to higher altitudes.

- Figure 8: How was the vertical level averaging done? The caption states "averaging all the vertical layers". The standard definition of atmospheric burden is the sum of the abundances in each layer (i.e. concentration times the height of the layer). If this is not

what is shown here, another word than burden should be used.

Response: Corrected. We have recreated the Figure. We now summed the vertical levels in order to be consistent with the definition of "burden" as pointed by the reviewer. Note that the scale has now changed to include the much higher values resulting from the summing of the layers.

Please also note the supplement to this comment:
http://www.atmos-chem-phys-discuss.net/acp-2015-994/acp-2015-994-AC3-supplement.pdf
* * *
[Figure]

**Fig. 1.**

Z (mb)

MMR[Y=60N:80N@AV4]
MMR[Y=20S:20N@AV4]
MMR[Y=60N:80N@AV4]
MMR[Y=20N:60N@AV4]
MMR[Y=60S:20S@AV4]
MMR[Y=67S:60S@AV4]

**Fig. 2.**